# DRAGFLOW: UNLEASHING DiT PRIORS WITH REGION BASED SUPERVISION FOR DRAG EDITING

**Zihan Zhou**[1,*], **Shilin Lu**[1,*], **Shuli Leng**[1], **Shaocong Zhang**[1],
**Zhuming Lian**[1], **Xinlei Yu**[2], **Adams Wai-Kin Kong**[1]
[1]Nanyang Technological University, [2]National University of Singapore
{zihan010, shilin002, nie25.ls3409, zhan0711, zhuming001}@e.ntu.edu.sg
xinlei.yu@u.nus.edu  adamskong@ntu.edu.sg

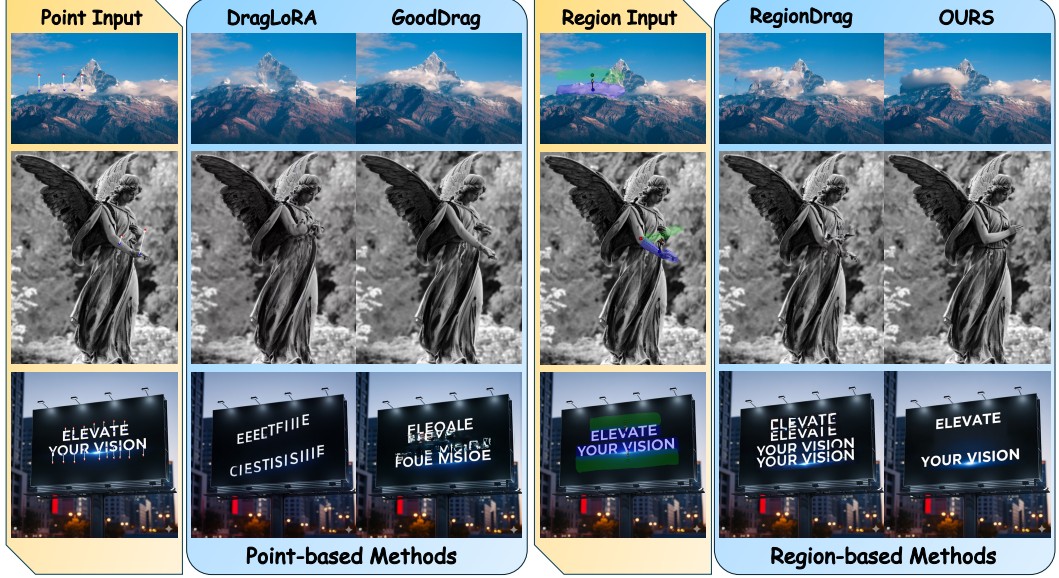

Figure 1: Comparison of drag-editing results between baselines and our method, **DragFlow**. DragFlow successfully unleashes FLUX's stronger generative prior, removing the distortions that previous methods produced on challenging scenarios.

## ABSTRACT

Drag-based image editing has long suffered from distortions in the target region, largely because the priors of earlier base models, Stable Diffusion, are insufficient to project optimized latents back onto the natural image manifold. With the shift from UNet-based DDPMs to more scalable DiT with flow matching (e.g., SD3.5, FLUX), generative priors have become significantly stronger, enabling advances across diverse editing tasks. However, drag-based editing has yet to benefit from these stronger priors. This work introduces DragFlow, the first framework to effectively harness FLUX's rich prior via region-based supervision, enabling full use of its finer-grained, spatially precise features for drag-based editing and achieving substantial improvements over existing baselines. We first show that directly applying point-based drag editing to DiTs performs poorly: unlike the highly compressed features of UNets, DiT features are insufficiently structured to provide reliable guidance for point-wise motion supervision. To overcome this limitation, DragFlow introduces a region-based editing paradigm, where affine transformations enable richer and more consistent feature supervision. Additionally, we integrate pretrained open-domain personalization adapters (e.g., IP-Adapter) to enhance subject consistency, while preserving background fidelity through gradient mask-based hard constraints. Multimodal large language models (MLLMs) are further employed to resolve task ambiguities. For evaluation, we curate a novel Region-based Dragging benchmark (ReD Bench) featuring region-level dragging

---

*Equal Contribution.

instructions. Extensive experiments on DragBench-DR and ReD Bench show that DragFlow surpasses both point-based and region-based baselines, setting a new state-of-the-art in drag-based image editing. Code and dataset are available at https://github.com/Edennnnnnnnnn/DragFlow.

# 1 INTRODUCTION

Text-driven image editing (Labs et al., 2025) has made impressive progress, but natural language often underspecifies geometry and locality, leading to unintended changes. Drag-based image editing (Pan et al., 2023; Jiang et al., 2025; Xia et al., 2025) bridges this gap by enabling users to specify finer-grained, spatially localized motions through interactive drag instructions, yielding more controllable edits. Despite their success, however, these methods often introduce unnatural deformations and distortions, especially in images with intricate details or complex structures.

We attribute this limitation to the insufficient generative prior of Stable Diffusion (SD) (Rombach et al., 2022a), the predominant base model, which struggles to constrain optimized latents back onto the natural image manifold. Past findings align with this view: applying nearly identical loss functions for drag editing yields far fewer unnatural distortions when using GAN-based priors compared to SD (Shi et al., 2024b). In parallel, recent advances in generative modeling have shifted from U-Net-based DDPMs to more scalable Diffusion Transformers (DiTs) (Peebles & Xie, 2023) trained with flow matching (Lipman et al., 2022) (e.g., SD 3.5 (Esser et al., 2024a), FLUX.1-dev (Black Forest Labs, 2024)), yielding substantially stronger priors that have propelled progress across various editing tasks (Lu et al., 2025; Yan et al., 2025; Wei et al., 2025; Deng et al., 2024; Wang et al., 2024; Rout et al., 2024). Yet, drag-based editing has not capitalized on these enhanced priors.

In this work, we pioneer the exploration of leveraging a stronger generative prior for drag editing. We first observe that directly applying previous drag editing methods to DiTs yields suboptimal results. Through a detailed analysis of features extracted from U-Nets and DiTs, we identify two core obstacles. First, point-based objectives used by prior drag methods mismatch DiT representations. U-Net bottlenecks produce spatially compact, highly compressed features that aggregate high-level semantics over broad receptive fields; supervising a single feature-map location therefore carries strong semantic evidence. DiTs, in contrast, yield finer-grained, spatially precise features with narrower receptive fields. Directly applying point-wise motion or tracking losses to DiTs provides weak semantic supervision and degrades editing effectiveness. Second, modern DiT models like FLUX are classifier-free-guidance (CFG)–distilled, which exacerbates inversion drift. As a result, standard key-value (KV) injection is insufficient to preserve subject identity consistency during drag edits.

To harness the potent priors of DiT-based models for drag-based editing, we introduce DragFlow, a novel region-based editing framework. DragFlow departs from point supervision and rethinks inversion and background handling to align with DiT feature geometry and the realities of CFG-distilled models. DragFlow advances the state of the art through three key innovations: (i) region-level motion supervision, which delivers richer and more consistent feature guidance via affine transformations; (ii) a replacement of traditional background consistency losses with hard constraints that preserve the background while updating only the editable region; and (iii) adapter-enhanced inversion, which injects subject representations from a pretrained open-domain personalization adapter (e.g., IP-Adapter (Ye et al., 2023)) into the base model's prior, achieving markedly superior subject fidelity under edits. Together, these components make drag-based editing practical with DiT backbones: they harness the stronger generative prior without sacrificing controllability, reducing deformation artifacts and improving faithfulness on complex, detail-rich images.

For evaluation, we introduce the Region-based Dragging Benchmark (ReD Bench). Each sample in ReD Bench is equipped with point-to-region alignment, explicit task tags spanning relocation, deformation, and rotation, and contextual descriptions that clarify user intent. We validate DragFlow extensively on both DragBench-DR (Lu et al., 2024a) and ReD Bench. Results demonstrate that DragFlow consistently outperforms state-of-the-art (SOTA) baselines.

## 2    RELATED WORK

Recent advances in diffusion models have led to a surge of interactive image-editing techniques, enabling users to intuitively reposition or deform specific regions of an image through drag-based interactions. Existing methods for drag-based editing can be broadly grouped into three categories.

**(i) Optimization-based methods** (Xia et al., 2025; Ling et al., 2024; Liu et al., 2024; Zhang et al., 2024c; Jiang et al., 2025; Shi et al., 2024b; Karras et al., 2022; Mou et al., 2023; 2024; Lin et al., 2025; Hou et al., 2024; Cui et al., 2024; Luo et al., 2024; Choi et al., 2024), which is the most prevalent category, iteratively refine inverted noisy latents during inference. These techniques are predominantly point-based: they accept point-wise drag instructions as input and employ motion supervision and point tracking, both of which operate at the point level. However, they often yield unnatural deformations or distortions in the edited images, primarily because the optimized latents deviate from the natural image manifold learned by the base model, residing in out-of-distribution regions. Thus, many studies have focused on more judicious optimization strategies to ensure that the resulting latents can be more readily mapped back to plausible natural images. **(ii) Finetuning-based methods** (Shin et al., 2024; Shi et al., 2024a), which adapt a base text-to-image (T2I) diffusion model using curated video datasets. Yet, the inherent mismatch between video data and the precise instructions required for drag editing, coupled with the scarcity of high-quality, large-scale training data, limits their generalization. These methods usually fail to achieve complete drag effects and are prone to distortions. **(iii) Methods that avoid both finetuning and optimization** (e.g., RegionDrag (Lu et al., 2024a) and FastDrag (Zhao et al., 2024d)), instead directly copying and pasting noisy latent patches to target locations computed via predefined mapping functions during inference. While this approach significantly enhances efficiency, it heavily relies on handcrafted priors for the mapping functions, often resulting in edited images that lack faithfulness and realism.

Our method falls within the optimization-based paradigm but innovates by replacing point-based motion supervision with region-level supervision, thereby enabling drag capabilities in DiTs. Like RegionDrag, our approach accepts region-based inputs; however, whereas RegionDrag requires users to manually predefine the target region mask (a challenging task in non-rigid scenarios), we only necessitate specifying a target point serving as the region's center. Moreover, RegionDrag performs point-wise copy-pasting within noisy latents, which, due to the handcrafted nature of its mappings, often fails to preserve internal region structures. In contrast, we treat the region as a cohesive unit, extracting holistic regional features to serve as supervision signals during latent optimization, thereby ensuring the integrity of internal structures.

## 3    METHODOLOGY

In this section, we begin by elucidating the inherent limitations of previous point-based drag editing frameworks when adapted to DiTs (Sec. 3.1). Building on this analysis, we introduce a region-level affine supervision strategy (Sec. 3.2). To further enhance fidelity, we incorporate hard-constrained background preservation (Sec. 3.3) and adapter-enhanced subject consistency mechanisms (Sec. 3.4), addressing inversion drifts in DiTs.

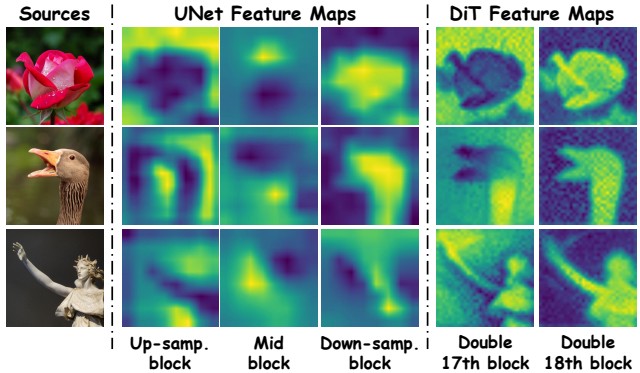

Figure 2: Comparison of feature maps extracted from UNet and DiT at the same denoising step. UNet produces spatially compact, highly compressed features that capture high-level semantic information, whereas DiT generates finer-grained, spatially precise representations.

### 3.1    WHY POINT-BASED DRAG FAILS ON DIT

To harness the robust prior of FLUX for drag-based image editing, we initially applied established point-based drag editing frameworks, including image inversion, motion supervision, point tracking, and key-value injection, directly to FLUX. Surprisingly, as shown in Fig. 6, this straightforward adaptation offers only limited improvements compared to its counterpart in SD.

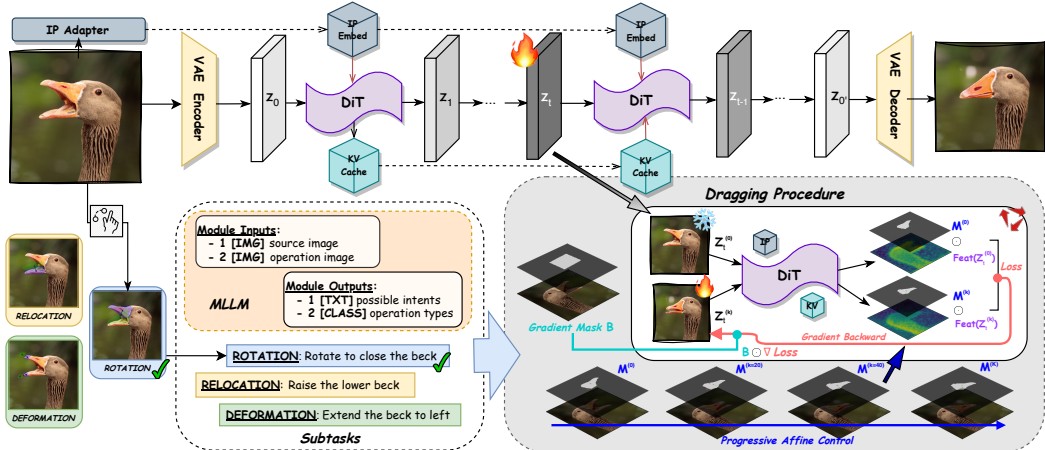

Figure 3: **Overview of the DragFlow framework.** The original image is inverted into a noisy latent space and iteratively optimized under the proposed region-level affine supervision. Subject consistency is reinforced through key-value (KV) injection and our adapter-enhanced inversion, while background fidelity is maintained via gradient mask-based hard constraints. In addition, a multimodal large language model (MLLM) is employed to better interpret and clarify user intents.

We attribute this performance gap to fundamental differences in the feature granularity extracted by the DiT and UNet, which significantly impact the effectiveness of point-wise motion supervision and tracking methods. As illustrated in Fig. 2, the UNet architecture, due to its bottleneck design, produces spatially compact and highly compressed features that encapsulate high-level semantic representations. In contrast, DiT generates finer-grained, spatially precise features. Consequently, in UNet, each point on the feature map aggregates semantic information from a broad receptive field in the input image, whereas in DiT, each point corresponds to a narrower region.

This difference directly impacts point-based methods, which rely on computing losses for motion supervision and point tracking using individual points on the feature map. In UNet, the broader receptive field of each feature point provides rich semantic context, enabling effective motion supervision and tracking. However, in DiT, the finer-grained features, with their narrower receptive fields, capture less semantic information per point, undermining the efficacy of these point-based methods when applied directly to DiT.

## 3.2 REGION-LEVEL AFFINE SUPERVISION

To leverage the powerful prior of FLUX for drag-based image editing, we introduce **DragFlow**, a region-based framework.

**User Input Specification.** In our framework, the user designates source region masks $\{\boldsymbol{M}_i\}_{i=1}^N$, each paired with a corresponding target point $\boldsymbol{t}_i = (x_i, y_i)$. The target point serves as the centroid of the target region. The expected target region mask can be obtained via an affine transformation (see Appendix C.1 for details). As illustrated in Fig. 3, we harness the capabilities of a multimodal large language model (MLLM), GPT-5 (OpenAI, 2025), to infer users' underlying intentions and thereby facilitate drag-based editing. The model receives as input the original image together with user-provided drag instructions (i.e., source region masks and target points). We then prompt the MLLM with carefully designed in-context examples, which guide it to produce two outputs: (i) a class label indicating the type of editing operation, and (ii) a textual description articulating the inferred editing intent (see Appendix D.3 for details on the prompting strategy). The class label determines which affine transformation is applied, while the textual description serves as a natural-language prompt for the generative model during the drag-editing process.

**Iterative Latent Optimization.** Given an input image $\boldsymbol{x}$, it is first encoded by the VAE to produce the latent $\boldsymbol{z}$, which is then inverted to obtain the noisy latent $\boldsymbol{z}_t$ where $t \in [0, T]$. We optimize $\boldsymbol{z}_t$ over $k$ iterations where $k \in [0, K]$, denoted as $\boldsymbol{z}_t^{(k)}$, such that subsequent denoising of $\boldsymbol{z}_t^{(k)}$ produces

an output image that fulfills the user-specified drag operations. This optimization is guided by the following loss function:

$$\mathcal{L}_{\text{Drag}} = \sum_{i=1}^{N} \gamma_i \cdot \left\| M_i^{(k)} \odot F\left(z_t^{(k)}\right) - \text{sg}\left[ M_i^{(0)} \odot F\left(z_t^{(0)}\right)\right] \right\|_1, \quad \text{where} \quad \sum_{i=1}^{N} \gamma_i = 1. \quad (1)$$

Here, $z_t^{(0)} \triangleq z_t$ is the initial unoptimized latent, while $z_t^{(k)}$ represents the latent after $k$ optimization iterations. The function $F(\cdot)$ extracts features from DiT, with the specific feature layers detailed in Appendix C.5. The operator $\text{sg}[\cdot]$ denotes stop-gradient. The weighting coefficient $\gamma_i$ balances multiple drag operations within the same image, determined adaptively according to the relative size of the corresponding manipulated regions (see Appendix D.1 for the formulation in details). Finally, $M_i^{(0)} \triangleq M_i$ is the user-provided mask for the source region, and $M_i^{(k)}$ specifies the corresponding target region, where we enforce similarity to the features of the source region.

**Affine Transformation for Mask Propagation.** The target mask $M_i^{(k)}$ is derived from the source mask $M_i^{(0)}$ via an affine transformation:

$$M_i^{(k)} = \Omega\left(M_i^{(0)}, \xi_i^{(k)}\right), \quad \xi_i^{(k)} = \begin{cases} \dfrac{k}{K}(t_i - b_i), & \text{(Relocation \& Deformation)} \\ \left(\dfrac{k}{K}\angle b_i a_i t_i, \ a_i\right), & \text{(Rotation)} \end{cases} \quad (2)$$

where $\Omega$ applies the affine transformation (affine computation detailed in Appendix C) to $M_i^{(0)}$ with parameters governed by $\xi_i^{(k)}$. Different drag types influence the affine matrix parameters distinctly: for relocation and deformation, these are determined by the vector from the target point $t_i$ to the centroid $b_i$ of the source region, where $b_i = (1/|M_i^{(0)}|) \sum_{q \in M_i^{(0)}} q$; for rotation, they are governed by the angle $\angle b_i a_i t_i$ formed by $t_i$, $b_i$, and the user-specified anchor point $a_i$. The linear schedule $k/K$ moves the mask smoothly from the source configuration toward the target over $K$ iterations.

**Why Region-level Supervision.** This formulation causes the target mask $M_i^{(k)}$ to translate (or rotate) progressively from the source centroid toward the target point as $k$ increases. Intuitively, it transports the object's features from the source region to the destination step by step. Although this echoes the high-level idea of point-based dragging, there are two crucial differences:

*Feature granularity.* Point-based methods compare features only at a handle point, either against its previous location or against the original source point. In contrast, we match features between entire source and target regions. Region-level supervision provides richer semantic context and mitigates myopic gradients, which leads to more effective latent updates on FLUX.

*Tracking requirement.* Point-based approaches require handle point tracking to keep the extracted features aligned with the moving content. Without tracking, naively advancing the handle along a straight line is brittle: even a slight deviation of the optimized content from that line causes subsequent local features to mismatch the intended structure, causing error accumulation and eventual drag failure. Our region-level supervision avoids this failure mode. Because we compare features over regions rather than at a single point, we do not need to pinpoint a feature extraction location on the object at each step. This makes the procedure substantially more stable and robust, eliminating the need for explicit tracking; it suffices to shift the source region mask along the path from the source centroid to the target point.

## 3.3 BACKGROUND PRESERVATION

Prior work typically enforces fidelity in non-editable regions via an auxiliary consistency loss:

$$\mathcal{L}_{\text{BG}} = \left\| \left(z_{t-1}^{(k)} - \text{sg}[z_{t-1}^{(0)}]\right) \odot (1 - B) \right\|_1, \quad (3)$$

where $B$ denotes the mask that specifies the editable region. In practice, this term competes with the feature-matching objective, making performance highly sensitive to its weight. The issue is exacerbated in FLUX, a CFG–distilled model that exhibits larger image-inversion drift than non-distilled counterparts. Compounding this, the consistency loss is evaluated against the inverted latent $z_{t-1}^{(0)}$,

which is treated as ground truth; when the inversion is biased, this target is unreliable, and the loss misguides optimization rather than helping it.

Instead of balancing competing losses, we hard constrain the background and update only the editable region:

$$z_t^{(k+1)} = \boldsymbol{B} \odot \left( z_t^{(k)} - \alpha \cdot \frac{\partial \mathcal{L}_{\text{Drag}}}{\partial z_t^{(k)}} \right) + (\boldsymbol{1} - \boldsymbol{B}) \odot z_t^{\text{orig}}, \tag{4}$$

where $\mathbf{B}$ denotes the mask that specifies the editable region (see Appendix D.2 for extraction details), and $z_t^{\text{orig}}$ is obtained from a pure reconstruction path. Implementationally, this requires an additional reconstruction branch that starts from the inverted latent $z_t$; the overhead is modest and, critically, it yields substantially better background preservation in challenging FLUX settings.

## 3.4 SUBJECT CONSISTENCY ENHANCEMENT

While our proposed region-level affine supervision enables drag-based editing using the prior of FLUX, it still suffers from subject inconsistency between the source and edited

Table 1: Inversion Performance (3,000 images).

| Method | LPIPS ↓ | SSIM ↑ | PSNR ↑ |
|---|---|---|---|
| DPM-Solver (Lu et al., 2022) Inv. (SD) | 0.167 | 0.799 | 26.31 |
| Fireflow Inv. w/o adapter (FLUX) | 0.283 | 0.703 | 20.43 |
| Fireflow Inv. w/ adapter (FLUX) | 0.173 | 0.784 | 25.87 |

images. A natural remedy is the KV injection, which is widely used when SD serves as the base model. In FLUX, however, KV injection underperforms, as shown in Fig. 4 (left). We attribute this gap to FLUX being a CFG-distilled model, which exhibits more pronounced inversion drift compared to non-distilled counterparts, as evidenced in Tab. 1.

To address this, we introduce adapter-enhanced inversion for DiT-based models. Specifically, pre-trained open-domain personalization adapters (e.g., IP-Adapter (Ye et al., 2023), PuLID (Guo et al., 2024), and InstantCharacter (Tao et al., 2025)) are trained to extract a subject's representation from a reference image, enabling its seamless integration into a T2I base model for generation across varied contexts. Leveraging this insight, we propose employing such adapters as auxiliary subject representation extractors. Without any additional fine-tuning, we inject the adapter's subject representation into the model prior, which substantially improves inversion quality (Tab. 1) and yields visibly better subject consistency under edits (Fig. 4 (right)).

## 4 EXPERIMENTS

### 4.1 IMPLEMENTATION DETAILS

We implement DragFlow using FLUX.1-dev (Black Forest Labs, 2024) as the base model. We adopt FireFlow (Deng et al., 2024) as the inversion algorithm for FLUX, employing 25 diffusion steps, with 6 steps skipped and drag editing commencing at the 19th step. We perform optimization at the 7th denoising step over 70 iterations, using a learning rate of 1000 for the first 50 iterations and 1200 for the final 20. The adapter employed is InstantCharacter (Tao et al., 2025). Additional implementation details are provided in Appendix C.5.

### 4.2 EXPERIMENTAL SETUP

**Benchmark.** To facilitate systematic evaluation of region-based image drag-editing methods, we introduce a new *Region-based Dragging Bench (ReD Bench)*. Existing datasets are often limited in scope. For example, *Drag-Bench* (Shi et al., 2024b) primarily provides

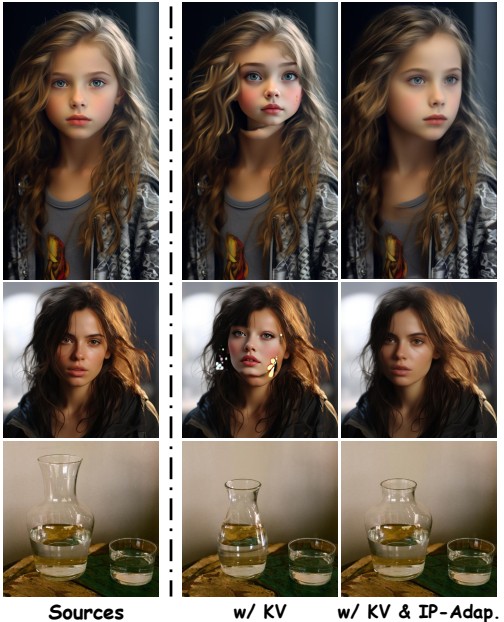

**Sources**    **w/ KV**    **w/ KV & IP-Adap.**

Figure 4: Visualization of the effect of adapter-enhanced inversion on subject consistency, compared with KV injection alone.

Table 2: Comparison of composition performance across two benchmarks. Optimal results are **bolded**, where the second-best own underlines. **Abbreviation:** OPT – optimization-based; FT – fine-tuning-based; NFT – neither fine-tuning nor optimization-based.

| Benchmark | Method | Category | #Params (M) | Image Fidelity | | | Mean Distance | |
|---|---|---|---|---|---|---|---|---|
| | | | | $\text{IF}_{bg} \uparrow$ | $\text{IF}_{s2t} \uparrow$ | $\text{IF}_{s2s} \downarrow$ | $\text{MD}_1 \downarrow$ | $\text{MD}_2 \downarrow$ |
| **ReD Bench** | RegionDrag (Lu et al., 2024a) | NFT | 0 | **1.000** | 0.957 | 0.957 | 33.69 | 6.38 |
| | FastDrag (Zhao et al., 2024d) | NFT | 0 | 0.928 | 0.950 | 0.941 | 23.37 | 5.00 |
| | InstantDrag (Shin et al., 2024) | FT | 914 | 0.930 | 0.949 | 0.946 | 24.38 | 4.54 |
| | DragLoRA (Xia et al., 2025) | OPT | 3.19 | 0.927 | 0.950 | 0.938 | 26.04 | 4.86 |
| | FreeDrag (Ling et al., 2024) | OPT | 0.07 | 0.941 | 0.947 | 0.956 | 30.31 | 6.08 |
| | DragNoise (Liu et al., 2024) | OPT | 0.33 | 0.942 | 0.932 | 0.975 | 45.46 | 8.85 |
| | GoodDrag (Zhang et al., 2024c) | OPT | 0.07 | 0.935 | 0.956 | 0.942 | 20.38 | 4.50 |
| | CLIPDrag (Jiang et al., 2025) | OPT | 0.07 | 0.952 | 0.942 | 0.965 | 33.84 | 6.98 |
| | DragDiffusion (Shi et al., 2024b) | OPT | 0.07 | 0.944 | 0.948 | 0.947 | 32.15 | 5.65 |
| | DragFlow (Ours) | OPT | 0 | 0.992 | **0.958** | **0.934** | **19.46** | **4.48** |
| **DragBench-DR** | RegionDrag (Lu et al., 2024a) | NFT | 0 | **1.000** | 0.942 | 0.960 | 32.32 | 6.31 |
| | FastDrag (Zhao et al., 2024d) | NFT | 0 | 0.938 | 0.947 | 0.952 | 35.96 | 6.60 |
| | InstantDrag (Shin et al., 2024) | FT | 914 | 0.944 | 0.945 | 0.966 | 36.26 | 6.99 |
| | DragLoRA (Xia et al., 2025) | OPT | 3.19 | 0.942 | 0.941 | 0.952 | 42.03 | 6.77 |
| | FreeDrag (Ling et al., 2024) | OPT | 0.07 | 0.955 | 0.946 | 0.967 | 34.77 | 6.81 |
| | DragNoise (Liu et al., 2024) | OPT | 0.33 | 0.956 | 0.943 | 0.977 | 39.31 | 7.69 |
| | GoodDrag (Zhang et al., 2024c) | OPT | 0.07 | 0.948 | 0.946 | 0.956 | 37.87 | 6.91 |
| | CLIPDrag (Jiang et al., 2025) | OPT | 0.07 | 0.962 | 0.945 | 0.972 | 38.06 | 7.45 |
| | DragDiffusion (Shi et al., 2024b) | OPT | 0.07 | 0.954 | 0.944 | 0.958 | 39.41 | 7.05 |
| | DragFlow (Ours) | OPT | 0 | 0.969 | **0.948** | **0.941** | **31.59** | **5.93** |

point-to-point guidance for dragging operations, which fails to capture the complexity of region-level manipulation. Extensions of DragBench with coarse region annotations also fall short, as they lack explicit point-to-region alignment and operation-specific instructions, such as task tags and anchor points.

In ReD, each sample not only includes point-level operations, but also offers the translated region-to-region correspondences, together with explicit task labels covering the three most common categories of dragging (i.e., relocation, deformation, and rotation). Moreover, ReD is enriched with detailed contextual descriptions and intent prompts. This richer supervision allows ReD to serve as a more reliable testbed for assessing our approach and comparing it with a range of SOTA baselines.

**Evaluation Metrics.** Following prior work (Liu et al., 2024; Zhang et al., 2024c; Shi et al., 2024b), we evaluate drag-based editing using a combination of *Mean Distance (MD)* and *Image Fidelity (IF)*. The standard MD metric (Shi et al., 2024b) quantifies the spatial correspondence of dragged content. We employ a masked variant, denoted as $\mathbf{MD}_1$, which computes correspondences only within the edited region, providing a more focused evaluation of alignment quality. In addition, we also adopt the variant proposed by Lu et al. (2024a), denoted as $\mathbf{MD}_2$. IF assesses visual consistency between the original and edited images via LPIPS (Zhang et al., 2018). We employ three variants for a fine-grained analysis: $\mathbf{IF}_{bg}$: LPIPS computed on non-edited regions, capturing how well background content is preserved. $\mathbf{IF}_{s2t}$: LPIPS between the original source region and the edited target region, indicating how faithfully source content is transferred. $\mathbf{IF}_{s2s}$: LPIPS between the original and edited source regions, measuring how effectively the source is cleared after transfer.

### 4.3 COMPARISON WITH BASELINES

**Quantitative Analysis.** As shown in Tab. 2, our method achieves the lowest MD across both benchmarks, demonstrating the strongest spatial correspondence between user instructions and the resulting drag operations. The superior performance on $\mathbf{IF}_{s2s}$ highlights the method's ability to deliver precise and reliable content manipulation with high structural consistency and completeness. Although our approach ranks second on $\mathbf{IF}_{bg}$, the margin is marginal and largely attributable to the inherent inversion limitations of the CFG-distilled model. Nevertheless, our gradient mask–based hard constraints ensure robust background integrity, allowing our method to outperform most baselines despite these limitations.

**Qualitative Analysis.** Fig. 5 presents side-by-side comparisons with representative baselines. Our method consistently produces edits that accurately follow the specified dragging operations while preserving global scene coherence. In contrast, RegionDrag and InstantDrag often introduce struc-

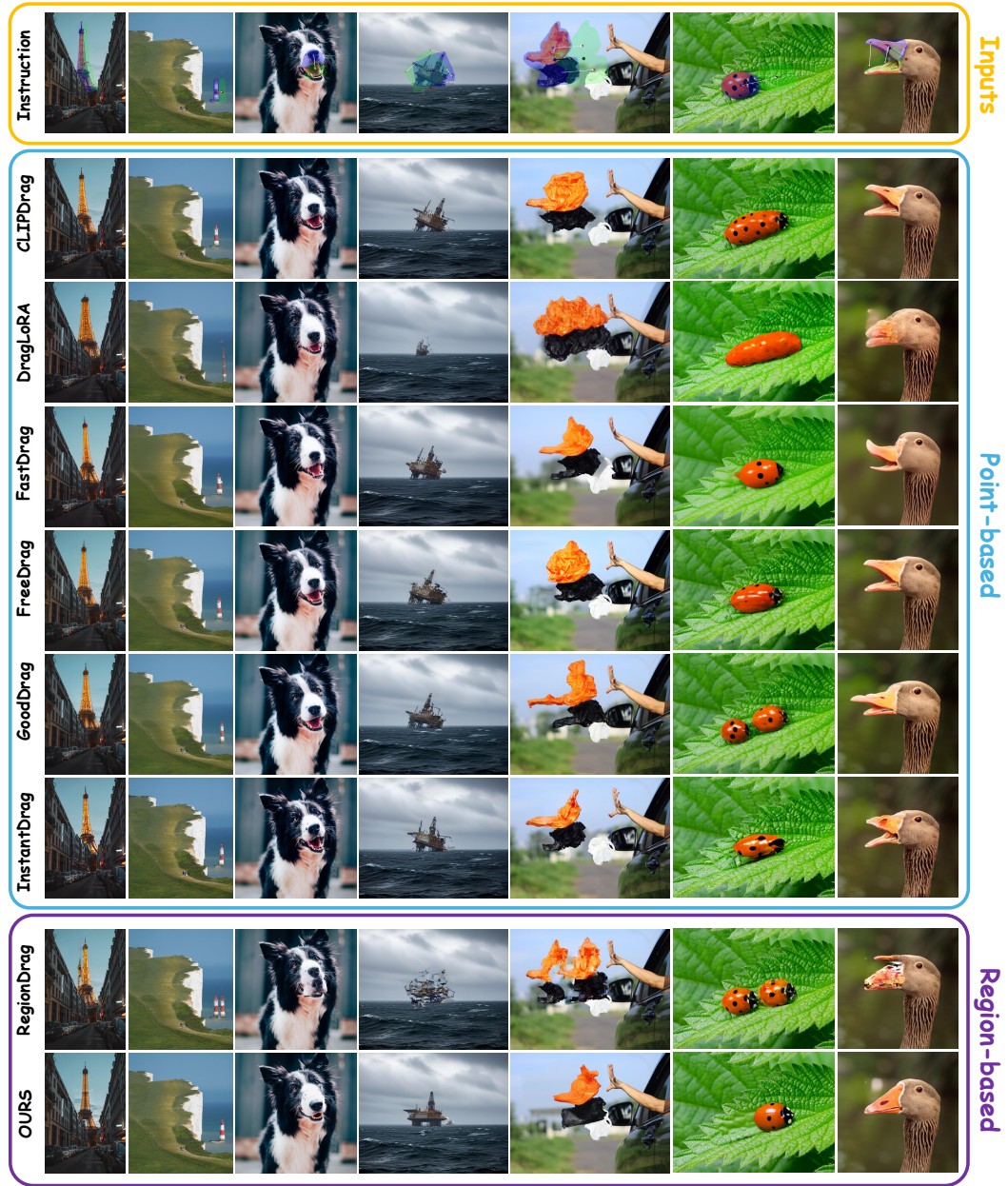

Figure 5: Qualitative comparison of our method with multiple baselines in challenging scenarios.

tural distortions, while FreeDrag and FastDrag struggle with complex transformations such as rotations. CLIPDrag and DragLoRA frequently misinterpret relocation as deformation, leading to unintended artifacts. Across all these scenarios, our approach demonstrates superior structural preservation, faithful intent alignment, and robust performance over a diverse range of editing tasks.

## 4.4 ABLATION STUDY

To assess the contribution of each component, we perform ablation studies (results shown in Fig. 6 and Tab. 3) by progressively incorporating modules into our framework. Specifically, we examine the impact of adopting **(a) region-based manipulation**, introducing **(b) mask-led background preservation**, and applying **(c) adapter-enhanced inversion**. In **(a)**, transitioning from points to region-based control leads to consistent gains across all evaluation metrics. Compared with the baseline, regional manipulation reduces $\mathbf{MD}_1$ by 19.95 and increases $\mathbf{IF}_{s2t}$ by 0.027, highlighting its ability to provide semantically richer and more coherent feature guidance. These results support our hypothesis that the regional affine strategy constitutes a more principled paradigm for modern generative

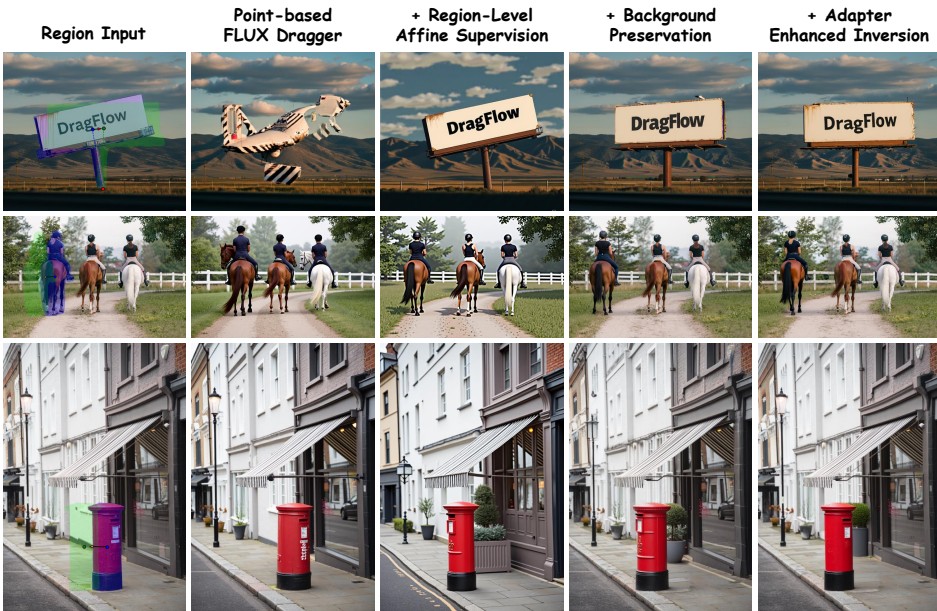

Figure 6: Qualitative ablation study comparing different variants of our framework.

Table 3: Ablation study on *ReD Bench* examining the impact of some key components.

| Configs | Image Fidelity | | | Mean Distance | |
|---|---|---|---|---|---|
| | $\mathbf{IF}_{bg} \uparrow$ | $\mathbf{IF}_{s2t} \uparrow$ | $\mathbf{IF}_{s2s} \downarrow$ | $\mathbf{MD}_1 \downarrow$ | $\mathbf{MD}_2 \downarrow$ |
| Baseline (Point-based FLUX) | 0.765 | 0.932 | 0.962 | 51.21 | 9.38 |
| + Region-Level Affine Supervision | 0.757 | 0.946 | **0.936** | 31.26 | 5.88 |
| + Background Preservation | 0.925 | 0.948 | 0.943 | 29.67 | 5.39 |
| + Adapter-Enhanced Inversion | **0.991** | **0.959** | 0.938 | **20.15** | **4.48** |

priors, effectively alleviating the intrinsic limitations of sparse point-level supervision. For **(b)**, our gradient mask design substantially improves background consistency, with $\mathbf{IF}_{bg}$ rising from $0.757$ to $0.925$, underscoring its effectiveness in preserving global backdrop integrity. Finally, the adapter-enhanced inversion in **(c)** significantly strengthens subject fidelity, as reflected by an increase in $\mathbf{IF}_{s2t}$ from $0.948$ to $0.959$, thereby confirming its capability to maintain foreground consistency under drag-editing. Taken together, these results clearly demonstrate that each component is effective on its own while synergizing to validate the overall design of our method.

## 5 CONCLUSION

We propose DragFlow, the first drag-based image editing framework tailored for DiTs. By rethinking supervision, inversion, and background handling in light of the unique representational and training properties of DiTs, DragFlow unlocks their powerful generative priors for controllable, fine-grained drag editing. Our three contributions, namely region-level motion supervision, background hard constraints, and adapter-enhanced inversion, collectively address the weaknesses of prior drag approaches, mitigating deformation artifacts while preserving subject identity and image realism. To support rigorous evaluation, we introduce ReD Bench, a benchmark designed around region-aware annotations and explicit task categories. Across both ReD Bench and DragBench-DR, DragFlow consistently surpasses existing state-of-the-art methods, demonstrating stronger faithfulness, better controllability, and higher-quality outputs.

**Limitations and Future Work.** Since the FLUX model we employ for drag-based editing is a CFG-distilled variant, its inversion drift is notably larger than that of non-distilled counterparts. Although we mitigate this issue through adapter-enhanced inversion, images with highly intricate structures still exhibit detail loss in the reconstruction. Consequently, the drag-editing results inherit these artifacts, leading to degraded visual quality. Future research could benefit from techniques

or advanced adapter architectures that further strengthen inversion fidelity, thereby reducing such artifacts and enhancing overall editing performance.

## ACKNOWLEDGMENTS

This research is supported by the National Research Foundation, Prime Minister's Office, Singapore, and the Ministry of Digital Development and Information, under its Online Trust and Safety (OTS) Research Programme (MDDI-OTS-001). Any opinions, findings and conclusions or recommendations expressed in this material are those of the author(s) and do not reflect the views of National Research Foundation, Prime Minister's Office, Singapore, the Ministry of Digital Development and Information, or the Centre for Advanced Technologies in Online Safety.

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

# Appendix

## Table of Contents

## A  ADDITIONAL RELATED WORK

**Generative Model-based Image Editing.**  Recent and significant advancements in generative models (Chang et al., 2023; Ding et al., 2022; Nichol et al., 2021; Ramesh et al., 2022; Rombach et al., 2022a; Saharia et al., 2022; Yu et al., 2022; Peebles & Xie, 2023; Esser et al., 2024a) have enhanced many applications (Avrahami et al., 2023; Ruiz et al., 2023; Hertz et al., 2022; Kim et al., 2022; Tumanyan et al., 2023; Zhou et al., 2025; 2023; 2024a;b;c; Zhao et al., 2024a; 2025b;a; 2024b; 2026; Wang et al., 2025; Wang & Chen, 2025a;b; Lu et al., 2024b;c; Li et al., 2025; Ren et al., 2025; Gao et al., 2024; 2025; Zhu et al., 2024; 2025; Yu et al., 2025a;b). In this study, we focus on real image editing, which allows users to freely modify actual photographs, producing highly realistic results. Typically, the inputs for image editing include an image and various conditions that help users accurately describe their desired changes. These conditions can encompass text prompts using natural language to specify the edits (Brooks et al., 2023; Zhang et al., 2024b; Fu et al., 2023; Zhang et al., 2024a), region masks to designate areas for modification (Zhao et al., 2024c; Zhuang et al., 2023; Wang et al., 2023), additional images to provide desired styles or objects (Chen et al., 2024; Lu et al., 2023; Yang et al., 2023), and drag points (Lu & Han, 2025; Nie et al., 2024) that enable users to interactively move specific points in the image to target positions.

## B  PRELIMINARY INFORMATION

This section provides essential background knowledge on diffusion models and point-based image drag-editing techniques. We begin by discussing the evolution of diffusion methods, tracing the transition from early frameworks of *Stable Diffusion (SD)* Series with *Denoising Diffusion Implicit Models (DDIM)* (Song et al., 2022) to more advanced rectified flow models and ODE solutions. These advancements enhance the generative process, facilitating more efficient and deterministic sampling. In the latter part, we explore point-based drag-editing methods, highlighting how they allow users to manipulate key points for image modification directly. This approach contrasts with region-based methods by leveraging specific user-given control points for supervision and tracking.

### B.1  DIFFUSION METHODS

Recent advances in text-to-image generative models have driven a rapid evolution in architectural paradigms, providing a strong impetus for advancing existing image editing methods. Innovations in newer architectures and learning objectives enrich prior knowledge and exhibit significantly improved capacity in data comprehension and semantic alignment. Early UNet–based diffusion frameworks, such as SD 1.5 and SD 2 from Rombach et al. (2022b), have been increasingly supplanted by DiT-based rectified flow designs with more robust pre-trained priors (e.g., FLUX.1 (Black Forest Labs, 2024) and SD 3 (Esser et al., 2024b)).

**Stable Diffusion with DDIM Inversion.**  Early drag-based image editing techniques primarily use SD as the foundational framework. They leverage SD's UNet architecture for noise prediction and rely on DDIM inversion to derive noisy latents $z_t$ from the clean latent encoding $z_0$ of an input image $x$. DDIM operates through two core processes: a *forward inversion process* that iteratively adds noise to transform clean latents into noisy ones, and a *backward denoising process* that removes noise to generate edited outputs.

To formalize these processes, we first define $\bar{\alpha}_t = \prod_{s=1}^{t}(1 - \beta_s)$, where $\beta_s$ is the noise schedule at step $s$, and $t \in [0, T]$ indexes the diffusion steps, with $t = 0$ representing the clean state and $t = T$ the fully noisy state. Starting from the clean latent $z_0$ (i.e., the latent output of the VAE encoder), the forward inversion process computes $z_t$ from $z_{t-1}$ by first using the UNet noise predictor $\epsilon_\theta(\cdot, \cdot)$ to estimate the noise in $z_{t-1}$, which reconstructs an approximation of the original clean latent $\hat{z}_0$. This estimated clean latent is then perturbed to produce the next noisy latent $z_t$:

$$z_t = \sqrt{\bar{\alpha}_t} \cdot \hat{z}_0 + \sqrt{1 - \bar{\alpha}_t} \cdot \epsilon_\theta(z_{t-1}, t-1), \tag{5}$$

where the term $\sqrt{\bar{\alpha}_t} \cdot \hat{z}_0$ retains a scaled version of the estimated clean signal, and $\sqrt{1 - \bar{\alpha}_t} \cdot \epsilon_\theta(\cdot)$ adds controlled noise with scaling factors aligning with the pre-defined schedule $\bar{\alpha}_t$.

Oppositely, the backward denoising process iteratively refines $z_t$ to $z_{t-1}$, gradually reducing noise. From the current noisy latent $z_t$, the model first estimates the clean latent $\hat{z}_0$ by inverting the noise

addition—subtracting the predicted noise $\epsilon_\theta(\boldsymbol{z}_t, t)$ (scaled by the schedule $\sqrt{1-\bar{\alpha}_t}$):

$$\hat{\boldsymbol{z}}_0 = \frac{\boldsymbol{z}_t - \sqrt{1-\bar{\alpha}_t} \cdot \epsilon_\theta(\boldsymbol{z}_t, t)}{\sqrt{\bar{\alpha}_t}}. \tag{6}$$

Using this estimated clean latent $\hat{\boldsymbol{z}}_0$, the denoised latent $\boldsymbol{z}_{t-1}$ is computed by re-adding a controlled amount of noise to $\hat{\boldsymbol{z}}_0$, with the noise level reduced compared to $\boldsymbol{z}_t$, such as

$$\boldsymbol{z}_{t-1} = \sqrt{\bar{\alpha}_{t-1}} \cdot \hat{\boldsymbol{z}}_0 + \sqrt{1-\bar{\alpha}_{t-1}} \cdot \epsilon_\theta(\boldsymbol{z}_t, t). \tag{7}$$

This re-noising step ensures a gradual transition toward the clean state: as $t$ decreases, $\bar{\alpha}_{t-1}$ increases to approaching 1, so the weight on the estimated clean structure $\hat{\boldsymbol{z}}_0$ grows, while the weight on the noise term shrinks. This stepwise denoising is critical for editing, as the iterative formulation enables conditional guidance (e.g., text prompts or drag constraints) to be seamlessly injected at each stage, thereby ensuring precise and progressively refined control over the final output.

**Rectified Flow with ODE Solver.** In contrast, rectified flow models, as adopted in our DragFlow framework leveraging the DiT prior, reformulate the generative process through approximate straight-line trajectories between data and noise distributions, enabling more efficient and deterministic sampling with fewer steps. For simplicity, we normalize the time parameter to $t \in [0, 1]$.

The forward process linearly interpolates the latent as

$$\boldsymbol{z}_t = (1-t)\boldsymbol{z}_0 + t\boldsymbol{\epsilon}, \tag{8}$$

where $\boldsymbol{z}_0$ is the clean VAE-encoded latent (same as the UNet's), and $\boldsymbol{\epsilon} \sim \mathcal{N}(0, I)$ is standard Gaussian noise. The associated velocity field is constant, defined as $\frac{d\boldsymbol{z}_t}{dt} = \boldsymbol{\epsilon} - \boldsymbol{z}_0$. In this way, the model can model a parameterized velocity $v_\theta(\boldsymbol{z}_t, t) \approx \boldsymbol{\epsilon} - \boldsymbol{z}_0$ via objectives like conditional flow matching, by minimizing

$$\mathcal{L} = \mathbb{E}_{t, \boldsymbol{z}_t, \boldsymbol{\epsilon}} \left\| v_\theta(\boldsymbol{z}_t, t) - (\boldsymbol{\epsilon} - \boldsymbol{z}_0) \right\|_2^2. \tag{9}$$

For the backward denoising process, we solve the ordinary differential equation (ODE) $d\boldsymbol{z}/dt = v_\theta(\boldsymbol{z}, t)$ backward in time from $t = 1$ to $t = 0$, starting from $\boldsymbol{z}_1 \approx \boldsymbol{\epsilon}$. In discrete steps (e.g., via *Fireflow Solver* (Deng et al., 2024)), this approximates:

$$\boldsymbol{z}_{t-\Delta t} = \boldsymbol{z}_t + (-\Delta t) \cdot v_\theta(\boldsymbol{z}_t, t), \tag{10}$$

where $\Delta t > 0$ is the step size. Conversely, to obtain the noisy latent $\boldsymbol{z}_t$ during inversion (from clean $\boldsymbol{z}_0$ at $t = 0$ to $\boldsymbol{z}_1$ at $t = 1$), we integrate the ODE forward:

$$\boldsymbol{z}_{t+\Delta t} = \boldsymbol{z}_t + \Delta t \cdot v_\theta(\boldsymbol{z}_t, t). \tag{11}$$

This straight-path formulation in rectified flow supports fewer function evaluations compared to the curved trajectories with more complicated noise schedules in DDIM, enabling our region-level affine supervision in DragFlow to leverage more robust priors for drag-based editing.

## B.2 POINT-BASED IMAGE DRAG-EDITING

Point-based drag editing was first introduced by DragGAN (Pan et al., 2023), as an interactive paradigm for image manipulation, enabling users to directly move key points on an image to achieve desired transformations. Unlike text-guided methods, which often struggle with ambiguity in complex scenes, point-based dragging encodes editing intentions through spatially localized control points. This approach aligns well with users' intuitive interaction patterns, providing a straightforward yet effective means of specifying editing goals.

**User Input.** Each editing task requires the following basic inputs:

- An original image $x$ (converted to the initial latent $z_0$ through VAE encoding).
- A set of handle points $\{\boldsymbol{h}_i\}_{i=1}^n$ indicating locations to be manipulated.
- Corresponding target points $\{\boldsymbol{t}_i\}_{i=1}^n$ representing desired positions after dragging.
- Mask $\boldsymbol{B}$ to protect or constrain regions that should remain unchanged.

**Core Components.** The workflow typically consists of two interconnected modules:

1. **Motion Supervision (MS).** MS is designed to ensure the model preserves image features while enforcing alignment between source and target points. MS computes losses based on:

   - **Alignment Loss:** Measures feature differences between patches around original handle points and patches around their current predicted locations.

   $$\mathcal{L}_{\text{align}} = \sum_i \sum_{\boldsymbol{p} \in \Omega(\boldsymbol{h}_i^0, r)} \sum_{\boldsymbol{q} \in \Omega(\boldsymbol{h}_i^k, r)} \left\| F_{\boldsymbol{q}}(\boldsymbol{z}_t^k) - \text{sg}(F_{\boldsymbol{p}}(\boldsymbol{z}_t^0)) \right\|_1, \tag{12}$$

   where $F(\cdot)$ extracts local features, $\Omega(, r)$ denotes a patch of radius $r$, and $\text{sg}(\cdot)$ is the stop-gradient operator. $\boldsymbol{q}$ and $\boldsymbol{p}$ define the source patch and the manipulated patches, respectively.

   - **Smoothness Loss:** Encourages gradual changes in the feature space.

   $$\mathcal{L}_{\text{smooth}} = \sum_i \sum_{q \in \Omega(\boldsymbol{h}_i^k, r)} \left\| F_{\boldsymbol{q}}(\boldsymbol{z}_t^k) - \text{sg}(F_{\boldsymbol{q}}(\boldsymbol{z}_t^k)) \right\|_1, \tag{13}$$

   - **Mask Loss:** Penalizes unintended modifications outside user-defined regions.

   $$\mathcal{L}_{mask} = \left\| (\boldsymbol{z}_t^k - \text{sg}(\boldsymbol{z}_t^0)) \odot (1 - \boldsymbol{B}) \right\|_1, \tag{14}$$

   The overall motion supervision loss is:

   $$\mathcal{L}_{MS} = \beta \mathcal{L}_{align} + (1 - \beta) \mathcal{L}_{smooth} + \lambda \mathcal{L}_{mask}, \tag{15}$$

   which is backpropagated to iteratively update the latent code:

   $$\boldsymbol{z}_t^{k+1} = \boldsymbol{z}_t^k - lr \cdot \frac{\partial \mathcal{L}_{MS}}{\partial \boldsymbol{z}_t^k}. \tag{16}$$

2. **Point Tracking (PT).** PT updates handle point locations across diffusion steps to ensure they follow intended trajectories. For each handle point $\boldsymbol{h}_i^k$, the new location is determined via nearest-neighbor feature matching:

   $$\boldsymbol{h}_i^{k+1} = argmin_{q \in \Omega(\boldsymbol{h}_i^k, r_2)} \| F_q(\boldsymbol{z}_t^{k+1}) - F_{\boldsymbol{h}_i^0}(\boldsymbol{z}_t^0) \|_1. \tag{17}$$

**Workflow Summary.** The point-based editing process proceeds iteratively:

1. Apply several MS steps to align features toward target positions while preserving image consistency during the updating processes.
2. Perform PT to update handle points based on feature tracking.
3. Repeat the MS and PT cycle until the handle points converge to their targets.

To sum up, classic point-based drag editing leverages feature-level supervision and PT to enable localized manipulations, ensuring that the edited image remains coherent with the original while reflecting user-specified modifications.

However, this pipeline inherently suffers from several limitations: nearest-neighbor search and PT introduce high uncertainty during optimization; the explicit influence of each control point is confined to a narrow feature neighborhood, which restricts the scope of effective guidance. By contrast, region-based dragging naturally extends these ideas by operating over semantically coherent masks, thereby offering more stable, interpretable, and semantically meaningful editing outcomes.

## C  IMPLEMENTATION DETAILS

This section highlights the core motion strategy of region-based drag operations through progressive affine transformations. We define multiple types of drag operations, each designed to enhance user control and precision in image editing. Subsequent subsections detail the implementations of these operations, all leveraging progressively interpolated affine transformations for seamless transitions. Additionally, the final subsection offers supplementary settings related to the experimental section.

## C.1 Progressive Transformation in Subtasks

**Operation in Subtasks.** As introduced in the main text, we define **three** types of drag operations: *relocation*, local *deformation*, and *rotation*. Recall these definitions, the source masks $\boldsymbol{M}_i$ denote the initial regions, with centroids $\boldsymbol{b}_i$ guiding relocation and local deformation toward the target region indicated by the centroid $\boldsymbol{t}_i$. For rotation, the anchor $\boldsymbol{a}_i$ specifies the pivot. Each transformation is realized through affine updates, with parameters $\boldsymbol{\xi}_i^{(k)}$ interpolated over $K$ iterations to gradually propagate the source mask toward its target configuration. In the following subsections, we detail how these subtask settings facilitate the affine transformation in region-based image dragging.

**Progressive Motion Schedule.** In practice, for each region-specific drag operation, we only compute the full motion schedule $\boldsymbol{\xi}_i^{(K)}$ during the initial update iteration:

$$\boldsymbol{\xi}_i^{(K)} = \begin{cases} (\boldsymbol{t}_i - \boldsymbol{b}_i), & \text{(Relocation \& Deformation)} \\ (\angle \boldsymbol{b}_i \boldsymbol{a}_i \boldsymbol{t}_i, \ \boldsymbol{a}_i), & \text{(Rotation)}, \end{cases} \tag{18}$$

where $\boldsymbol{b}_i$, $\boldsymbol{t}_i$, and $\boldsymbol{a}_i$ denote the begin, target, and anchor points, respectively. Rather than recomputing the dragged and left distances at every iteration, subsequent steps obtain their progressive schedules directly via a linear interpolation:

$$\boldsymbol{\xi}_i^{(k)} = \tfrac{k}{K} \cdot \boldsymbol{\xi}_i^{(K)}, \quad \tfrac{k}{K} \in [0, 1]. \tag{19}$$

This design ensures that the motion evolves smoothly from the initial to the target state. The resulting progressive motion schedule is then injected into the affine transformation operator $\Omega(\cdot, \boldsymbol{\xi}_i^{(k)})$, to enforce consistent supervision over the sequence of incremental updates.

## C.2 Relocation Tasks

Relocation involves shifting an entire region to a new position while preserving its original geometry and scale. Recall our definition in Subsec. 3.2 This operation is parameterized by a displacement vector derived from the difference between the target point $\boldsymbol{t}_i$ and the source centroid $\boldsymbol{b}_i$, scaled linearly as $\boldsymbol{\xi}_i^{(k)} = \tfrac{k}{K}(\boldsymbol{t}_i - \boldsymbol{b}_i)$.

To apply this, for example, we can consider a point $\boldsymbol{p} = (u, v)$ within the source region mask $\boldsymbol{M}_i^{(0)}$. The transformed point $\boldsymbol{p}' = (u', v')$ can be computed via homogeneous coordinates:

$$\begin{bmatrix} u' \\ v' \\ 1 \end{bmatrix} = \begin{bmatrix} 1 & 0 & d_u \\ 0 & 1 & d_v \\ 0 & 0 & 1 \end{bmatrix} \begin{bmatrix} u \\ v \\ 1 \end{bmatrix}, \tag{20}$$

where $(d_u, d_v)$ is the displacement vector from $\boldsymbol{\xi}_i^{(k)}$. Breaking down the matrix:

- The top-left $2 \times 2$ submatrix is the identity, ensuring no rotation or scaling;
- The relocation components $d_u$ and $d_v$ in the first two rows of the third column directly add to the coordinates: $u' = u + d_u$, $v' = v + d_v$;
- The bottom row maintains homogeneity.

This matrix is applied across the whole operation patch, resulting in an efficient shift of the entire region in dragging steps.

## C.3 Deformation Tasks

Deformation enables localized adjustments by selectively displacing subregions, effectively altering the overall shape without global rigidity. In our setup, this is treated similarly to relocation, utilizing the same affine transformation method as described in the relocation operation. The key difference lies in its application: it targets only the edge areas of the object to be edited based on scene requirements, achieving effects such as elongation or shortening.

This selective application distinguishes deformation from full-region relocation, allowing for intuitive shape modifications as visualized in our method's overview.

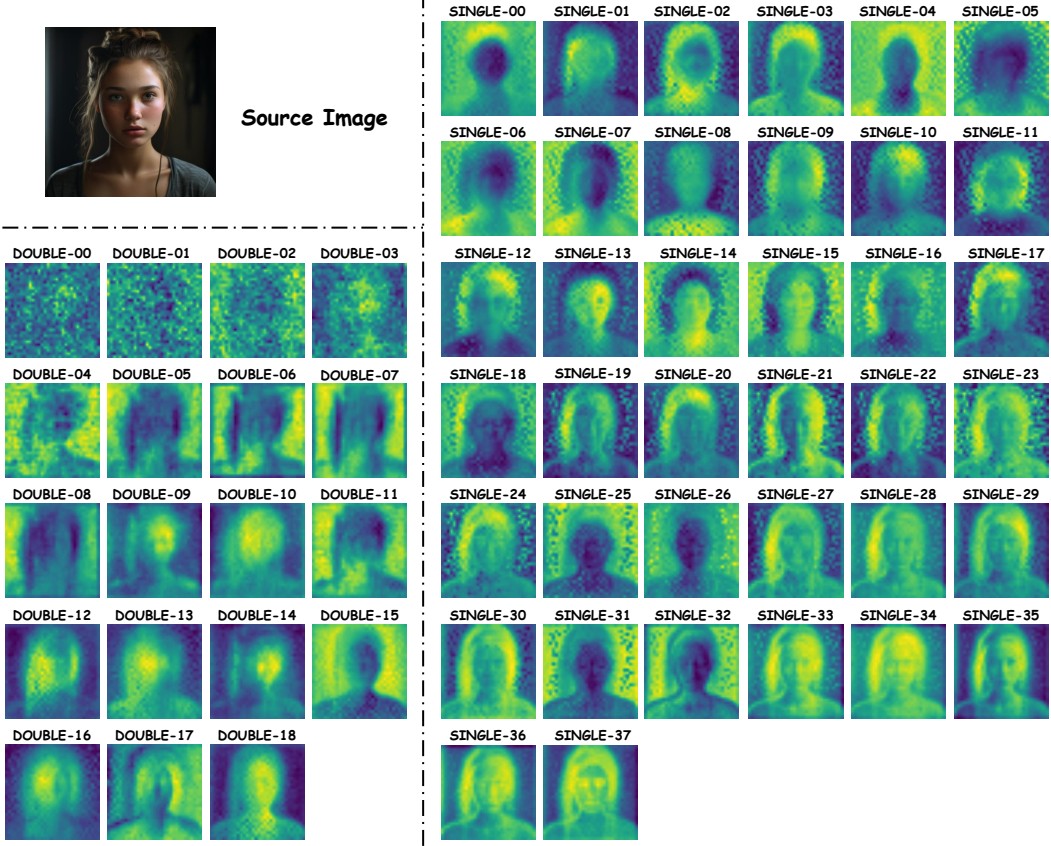

Figure 7: Visualization of DiT latent features (Sample 1 out of 2) based on PCA using the top 5 principal components.

## C.4 ROTATION TASKS

Rotation pivots a region around a fixed anchor, reorienting it by a progressively interpolated angle. The rotation angle is determined geometrically from the triangle formed by $\boldsymbol{b}_i$ (i.e., the source region centroid), $\boldsymbol{a}_i$ (i.e., the user-specified anchor), and $\boldsymbol{t}_i$ (i.e., the target region centroid). Over $K$ iterations, the interpolated parameters are defined as

$$\boldsymbol{\xi}_i^{(k)} = \left( \tfrac{k}{K} \angle \boldsymbol{b}_i \boldsymbol{a}_i \boldsymbol{t}_i, \ \boldsymbol{a}_i \right),$$

which guarantees a smooth progression from the original orientation toward the desired angular displacement.

For a point $\boldsymbol{r} = (w, z)$ in $\boldsymbol{M}_i^{(k)}$, rotated around anchor $\boldsymbol{c} = (w_c, z_c)$ by an angle $\phi$ derived from $\boldsymbol{\xi}_i^{(k)}$, the updated coordinates $\boldsymbol{r}' = (w', z')$ are obtained as

$$\begin{bmatrix} w' \\ z' \\ 1 \end{bmatrix} = \begin{bmatrix} 1 & 0 & w_c \\ 0 & 1 & z_c \\ 0 & 0 & 1 \end{bmatrix} \begin{bmatrix} \cos\phi & -\sin\phi & 0 \\ \sin\phi & \cos\phi & 0 \\ 0 & 0 & 1 \end{bmatrix} \begin{bmatrix} 1 & 0 & -w_c \\ 0 & 1 & -z_c \\ 0 & 0 & 1 \end{bmatrix} \begin{bmatrix} w \\ z \\ 1 \end{bmatrix}. \tag{21}$$

This composite transformation can be interpreted step by step:

- The rightmost matrix translates the point so that the anchor $\boldsymbol{a}_i$ coincides with the origin, removing bias from the global coordinate system.
- The central rotation matrix applies the angular displacement $\phi$, producing the intermediate rotated coordinates $(w'', z'')$ where $w'' = w\cos\phi - z\sin\phi$ and $z'' = w\sin\phi + z\cos\phi$.

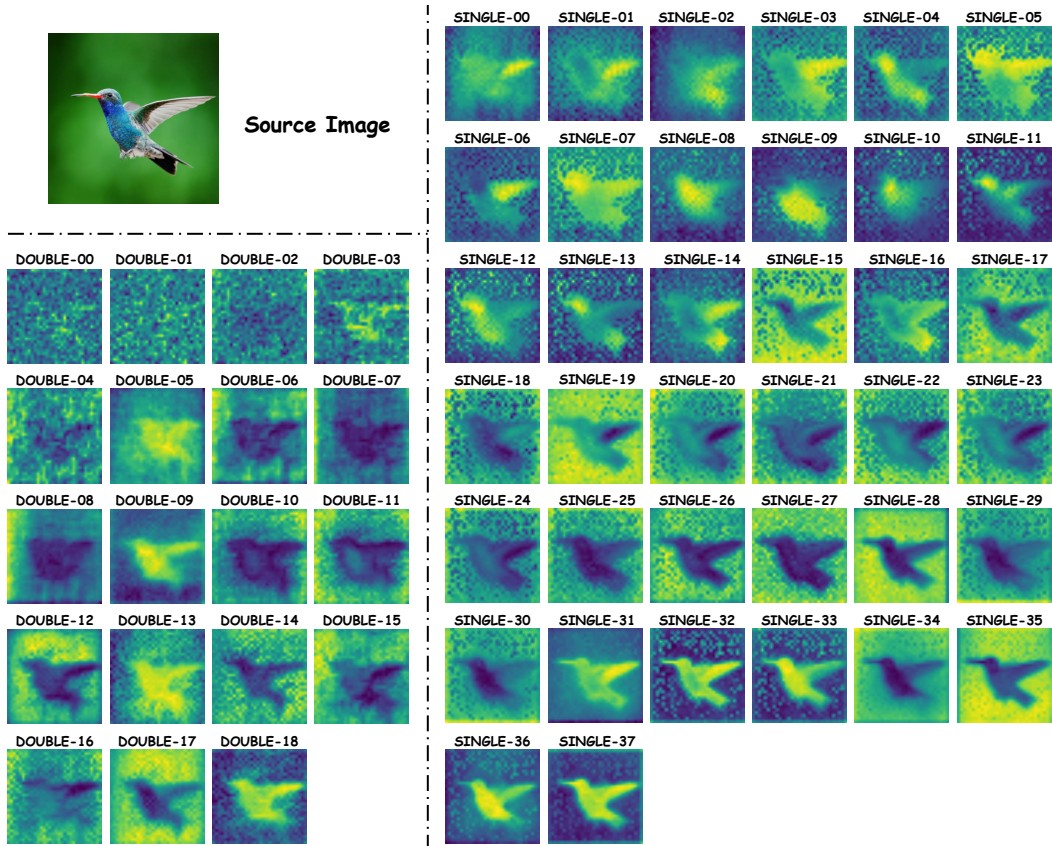

Figure 8: Visualization of DiT latent features (Sample 2 out of 2) based on PCA using the top 5 principal components.

- The leftmost matrix translates the rotated point back into the original coordinate frame, re-centering the result around $a_i$.

This decomposition not only clarifies the geometric intuition behind rotation but also integrates seamlessly with our iterative framework: at each step $k$, the patch masks $M_i^{(k)}$ are updated under this rotation transformation, ensuring gradual, controlled reorientation. Compared with relocation and deformation, rotation requires an explicit anchor to define the pivot, highlighting its distinct interaction design while still being governed by the same affine transformation principles.

## C.5 DETAILS ABOUT EXPERIMENTAL SETTINGS

**Layers for Feature Manipulation.** We empirically identify the **17th and 18th double-stream blocks** of the DiT backbone as the most representative features to facilitate drag optimization. To substantiate this choice, we conduct a visualization-based analysis of the main DiT modules (refer to Fig. 7 and Fig. 8 for details). Our selection is guided by the principle that effective feature blocks should retain high representational fidelity to the input. While certain blocks may encode large amounts of information, much of it can be dominated by noise. By examining PCA visualizations, we qualitatively assess whether the leading components align with the appearance of the original image, thereby confirming that the selected layers capture the essential visual characteristics required for precise and robust editing.

As illustrated in the figures, the latent representations from "DOUBLE-17" and "DOUBLE-18" retain rich and flexible features, which can be effectively manipulated during editing. In contrast, certain blocks (e.g., "SINGLE-20" and "SINGLE-30") exhibit overly "clean" latents, making it difficult to perform meaningful edits. Other blocks (e.g., "DOUBLE-04" and "SINGLE-06") con-

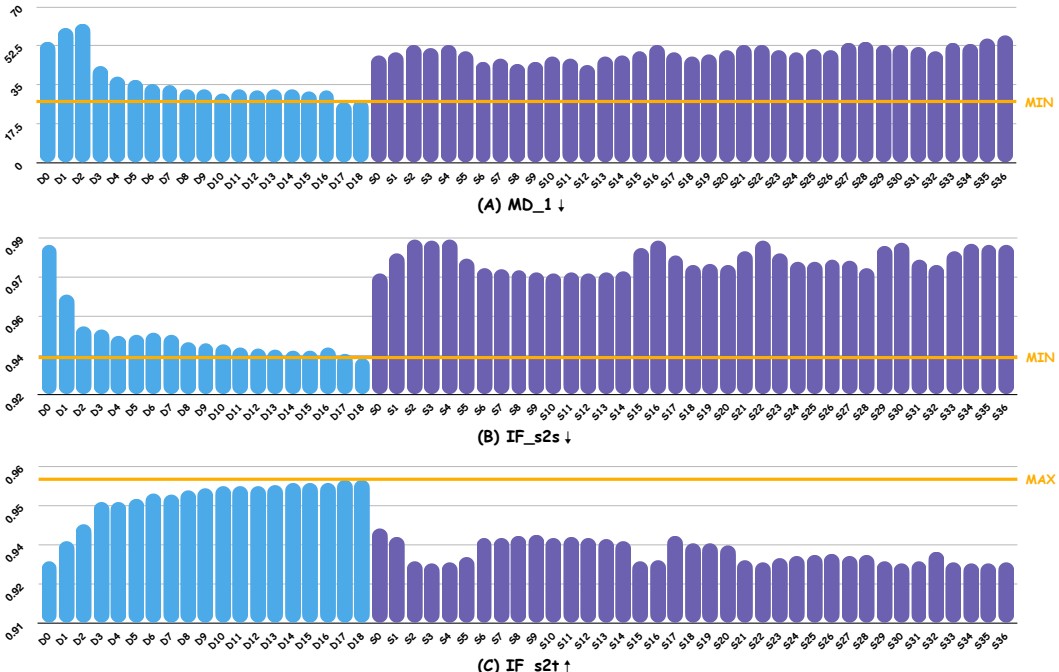

Figure 9: Comparison of dragging effectiveness across feature manipulation layers in the FLUX model. The analysis is evaluated on *ReD Bench*, demonstrating that **double-stream layers** (prefixed with "D", in blue) consistently outperform **single-stream layers** (prefixed with "S", in purple).

tain too few semantically informative features, where optimization tends to struggle in preserving source identity and achieving precise dragging control. Taken together, these comparisons suggest that our default setting, by using "DOUBLE-17" and "DOUBLE-18", offers a favorable balance: it preserves stable feature representations with sufficient semantic and spatial information, while also maintaining rich identity-related features to support effective editing. This justifies its adoption as the backbone choice in DragFlow.

**Layer-based Performance Comparison.** To further validate the effectiveness of our selected feature manipulation layers, we conducted an extensive ablation study focused on layer-wise feature selection. Specifically, under the same configuration, we compute the loss each time using the features from a particular layer. This procedure was repeated across all 57 layers from both the double-stream and single-stream branches of the FLUX model, enabling a systematic comparison of editing behaviors across various layers.

The results are summarized in Fig. 9. We evaluate three metrics of dragging effectiveness: $MD_1$, $IF_{s2s}$, and $IF_{s2t}$. As shown, operations performed on the double-stream layers (i.e., layers prefixed with "D") consistently outperform those on the single-stream layers (i.e., layers prefixed with "S") under the same dragging configuration. Notably, the 17th and 18th double-stream blocks (i.e., D17 and D18), which we ultimately adopt in our framework, exhibit the optimal performance robustly across all these metrics. This indicates that our chosen layers yield the highest-effectiveness solutions in terms of spatial displacement precision, preservation of post-dragging local features, and effective suppression of original-location features.

**Affine Transformation Steps.** During the dragging process, the motion from the source region to the target region is primarily executed over the first 50 steps ($k = 0$ to $49$), where each iteration progressively applies affine transformations to the latent representations, thereby facilitating the drag procedure through gradient updates. After this stage, DRAGFLOW performs an additional 20 optimization steps ($k = 50$ to $69$) by repeating the final affine transformation, which further refines the feature expression of the dragged object, helps it better adapt to the new semantic context, and enhances the consistency between the pre- and post-dragging regions.

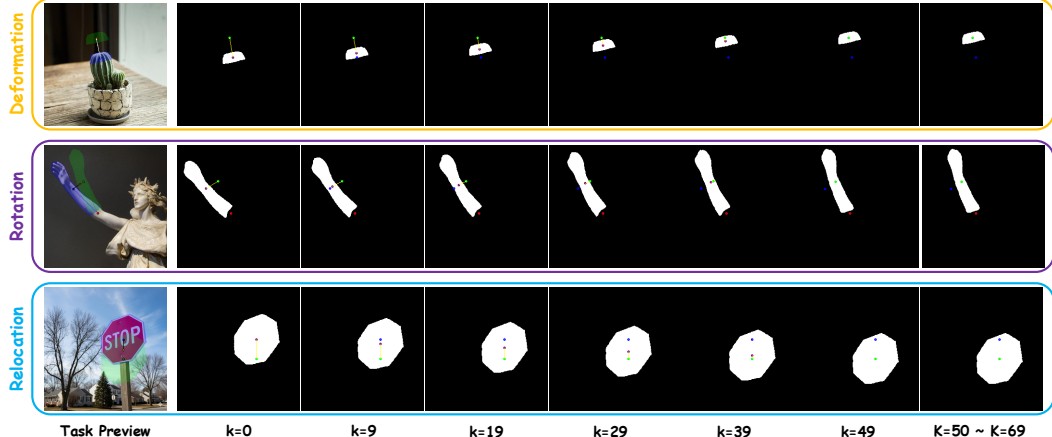

Figure 10: Region operation masks $\mathbf{M}_i^{(k)}$ created by progressive affine transformations at each step $k$ across multiple subtasks. Each dragging process consists of 50 steps ($k = 0$ to $49$), followed by 20 additional steps ($k = 50$ to $69$) that repeat the final motion iteration to further refine the feature quality of the post-dragging region.

As illustrated in Fig. 10, the patch operation mask $\mathbf{M}_i^{(k)}$, obtained through progressive transformations, evolves smoothly with increasing $k$, thereby driving the gradient optimization process to ensure effective dragging.

### C.6 Quantitative Analysis for DiT Supervision Granularity

Table 4: Quantitative comparison on *DragBench-DR* between Point-based and Region-based strategies for the DiT architecture. Both methods utilize the same backbone and auxiliary modules.

| Attempts | Image Fidelity | | | Mean Distance | |
|---|---|---|---|---|---|
| | $\mathbf{IF}_{bg} \uparrow$ | $\mathbf{IF}_{s2t} \uparrow$ | $\mathbf{IF}_{s2s} \downarrow$ | $\mathbf{MD}_1 \downarrow$ | $\mathbf{MD}_2 \downarrow$ |
| Point-based variant | 0.961 | 0.943 | 0.958 | 37.02 | 6.94 |
| Region-based variant (DragFlow) | 0.969 | 0.948 | 0.941 | 31.59 | 5.93 |

To further validate the analysis in Sec. 3.1, that DiTs' fine-grained features render sparse point supervision ineffective, we conducted a controlled quantitative comparison.

By evaluating two distinct supervision paradigms: a baseline *Point-based variant* (i.e., applying standard point to the DiT features) and our proposed *Region-based variant* (DragFlow). To ensure a rigorous evaluation of the supervision mechanisms in isolation, both variants were equipped with identical auxiliary modules, specifically Background Preservation and Adapter-Enhanced Inversion.

The comparative results are summarized in Tab. 4. These quantitative findings empirically corroborate our assertion that sparse point tracking is insufficient for the high-frequency feature landscape of DiTs, whereas region-level affine supervision effectively bridges this gap to fully unleash the model's generative potential.

## D Adaptive Input Processing

To ensure robust and user-aligned editing, DragFlow incorporates an adaptive input processing pipeline that systematically addresses three key challenges. First, when multiple operations are applied to a single image, we introduce an adaptive weighting scheme to balance region-level optimization and prevent dominance by large areas (see Appendix D.1). Second, to protect uneditable content during optimization, we design an adaptive gradient mask generation strategy that provides strict spatial constraints while maintaining flexibility for complex transformations (refer to Appendix D.2). Finally, to further reduce user effort and improve precision, we leverage a multimodal large language model (MLLM) to automatically generate candidate prompts and semantic tags, which are subsequently refined through minimal human intervention (detailed in Appendix D.3).

Together, these components form a cohesive input processing framework that enhances both the accuracy and practicality of DragFlow in real-world interactive editing scenarios.

## D.1 REGION WEIGHT REGULARIZATION FOR MULTI-OPERATIONS

When multiple drag operations are specified for a single image (i.e., $N > 1$), the loss function in Eq. 1 incorporates weighting coefficients $\{\gamma_i\}_{i=1}^N$ to balance the influence of each region, preventing larger regions from dominating the optimization. These weights are computed adaptively based on the relative sizes of the manipulated regions, ensuring equitable gradient contributions.

Formally, for each operation $i$, let $S_i$ denote the relative size of the source mask $\boldsymbol{M}_i^{(0)}$, defined as:

$$S_i = \frac{\sum \boldsymbol{M}_i^{(0)}}{|\boldsymbol{M}_i^{(0)}|}, \tag{22}$$

where $\sum \boldsymbol{M}_i^{(0)}$ is the number of non-zero (unmasked) pixels, and $|\boldsymbol{M}_i^{(0)}|$ is the total number of elements in the mask. Under this setup, a raw weight $w_i$ can then be calculated as:

$$w_i = \text{clamp}\left(1.0 + \frac{0.5}{S_i + 0.1}, 1.0, 5.0\right), \tag{23}$$

where $\text{clamp}(\cdot, a, b)$ restricts the value to the interval $[a, b]$. This formulation assigns higher weights to smaller regions to amplify their impact. And the total raw weight sum is $W = \sum_{i=1}^N w_i$. The normalized weights are:

$$\gamma_i = \begin{cases} \dfrac{1}{N} & \text{if } W = 0, \\[2mm] \dfrac{w_i}{W} & \text{otherwise.} \end{cases} \tag{24}$$

If $N = 1$, we set $\gamma_1 = 1$ directly. This adaptive scheme ensures $\sum_{i=1}^N \gamma_i = 1$, promoting balanced optimization across diverse region scales.

## D.2 ADAPTIVE GRADIENT MASK GENERATION

To sufficiently protect uneditable regions in our procedure, we generate an adaptive bounding mask $\boldsymbol{B}$ that only uncovers the edited areas. This mask $\boldsymbol{B}$ is derived from the user-provided source masks $\{\boldsymbol{M}_i^{(0)}\}_{i=1}^N$ and their target points $\{\boldsymbol{t}_i\}_{i=1}^N$, providing a static envelope for feature preservation during optimization. It differs from the iterative patch masks $\boldsymbol{M}_i^{(k)}$ by serving as a holistic boundary for the entire dragging process.

**Compare to Mask Loss.** Compared to the conventional mask loss strategy, gradient masking offers a more precise mechanism for constraining gradient flow, thereby preventing optimization from introducing unintended perturbations into uneditable regions. A key limitation of mask loss is that it can only partially restrict future updates, while inadvertently retaining undesired changes that have already been introduced. In contrast, gradient masking directly blocks gradient propagation to these regions, ensuring that they remain unaffected throughout the optimization process.

**Automatic Mask Creation.** For each source mask $\boldsymbol{M}_i^{(0)}$, we first apply the appropriate affine transformation (as defined in Eq. 3.2) with full interpolation (i.e., $k=K$) to obtain the final target mask $\boldsymbol{M}_i^{(K)} = \Omega(\boldsymbol{M}_i^{(0)}, \boldsymbol{\xi}_i^{(K)})$, where $\boldsymbol{\xi}_i^{(K)} = \boldsymbol{t}_i - \boldsymbol{b}_i$ for relocation/deformation or $(\angle \boldsymbol{b}_i \boldsymbol{a}_i \boldsymbol{t}_i, \boldsymbol{a}_i)$ for rotation.

Next, for each $i$, we compute the union mask $\boldsymbol{U}_i = \boldsymbol{M}_i^{(0)} \cup \boldsymbol{M}_i^{(K)}$. To ensure comprehensive coverage, we enclose $\boldsymbol{U}_i$ with a minimal rotated bounding rectangle. The points in $\boldsymbol{U}_i$ are collected as $\mathbf{P}_i = \{\mathbf{p} \mid \boldsymbol{U}_i(\mathbf{p}) = 1\}$. The rectangle parameters are then derived as

$$(\boldsymbol{c}_i, (w_i, h_i), \phi_i) = \text{minAreaRect}(\mathcal{P}_i), \tag{25}$$

where $\boldsymbol{c}_i$ is the center, $(w_i, h_i)$ are the dimensions, and $\phi_i$ is the rotation angle. The vertices of this rectangle are obtained via

$$\mathcal{V}_i = \text{boxPoints}((\boldsymbol{c}_i, (w_i, h_i), \phi_i)). \tag{26}$$

Since multiple operations may exist, each yielding an independent rectangle, we merge them into a single adaptive mask $\boldsymbol{B}$ by filling all convex polygons from an image-shape initiated canvas $\emptyset$ :

$$\boldsymbol{B} = \bigcup_{i=1}^{N} \text{fillConvexPoly}(\emptyset, \mathcal{V}_i). \tag{27}$$

Finally, $\boldsymbol{B}$ is binarized to $[0, 1]$, ensuring no excessive expansion while covering all transformed regions for effective integration into the latent optimization.

### D.3 LEVERAGING MLLM FOR PROMPT AND TAG GENERATION

In practical usage, the DragFlow framework provides a user-friendly interactive interface, enhanced by a multimodal large language model (MLLM) to support intuitive and precise image editing. The system emphasizes high automation to reduce user effort, while a two-step confirmation workflow ensures that user intentions are accurately communicated and ambiguities are minimized.

The interaction begins with the user providing an input image and roughly indicating the target region via a simple scribble, along with a click specifying the desired target position. The system converts the scribbled region into an initial operational mask representing the affected area. Leveraging an automatic mask generation algorithm (Appendix D.2), users do not need to redraw the target region for task specification. The original image and operational mask are then processed by the MLLM, which infers the user's editing intent and proposes a set of candidate prompts. Specifically, the MLLM generates ten potential prompts paired with corresponding task classes, from which users select the one that best reflects their intended operation.

After confirming the operational intent and task class, the interface produces a preview of the expected outcome based on the current operation parameters. For rotation operations, users specify an additional anchor point as the rotation center, which can be interactively adjusted while observing real-time updates in the preview. This iterative adjustment allows for fine-grained control, ensuring the final result closely aligns with user expectations. By combining automated inference with user-in-the-loop refinement, this design streamlines the editing process while faithfully translating high-level user intentions into precise image manipulations.

Here we present the prompt used for the prompt and tag generation procedure:

> *Refer to the original image, and the "dragged" image with the blue starting region, estimated green target region, and the arrow direction. You need to describe the content and the object for editing of the picture in English, in terms of "background details" and "editing changes". Then you should guess the editing intents from the user by selecting one label for each answer, where the label classes have {relocation, deformation, rotation}.*
>
> *Your tasks:*
>
> *- (a) You should first provide a detailed description about the original image (e.g., include, but are not limited to objects, spatial relationship, color, style, structure). Then try to describe the motion/editing in short words.*
> *- (b) You should provide the ten most possible guesses about the static condition of the after-dragged image, and at most 60 words for each. See if you can provide more details to facilitate the editing.*

### D.4 EFFECTIVENESS OF MLLM-DRIVEN INTENT PARSING

To quantitatively assess the contribution and robustness of the MLLM-driven intent parsing module, we conducted an extensive ablation study on the *DragBench-DR* dataset. This analysis compares the performance of DragFlow under three distinct experimental scenarios:

Table 5: To validate the effectiveness of prompts automatically generated by MLLM, we conducted an additional test. First, we examined whether using accurate and matching prompts is effective for drag operations. We considered **three** scenarios: *Null Prompt*, *Incorrect Prompt* (i.e., intentionally misinterpreted prompts), and *Matched Prompt* generated by MLLMs based on image and operation inputs. For the last scenario, we examine the effectiveness of **two** MLLMs (GPT-5 vs. QWen-VL). This experiment was conducted and scored on *DragBench-DR*.

| DragFlow w/ | Image Fidelity | | | Mean Distance | |
|---|---|---|---|---|---|
| | $\text{IF}_{bg} \uparrow$ | $\text{IF}_{s2t} \uparrow$ | $\text{IF}_{s2s} \downarrow$ | $\text{MD}_1 \downarrow$ | $\text{MD}_2 \downarrow$ |
| Null Prompt | 0.966 | 0.944 | 0.948 | 33.81 | 6.81 |
| Incorrect Prompt | 0.955 | 0.936 | 0.955 | 36.87 | 7.73 |
| Matched Prompt (by QWem-VL) | 0.968 | 0.947 | 0.945 | 32.42 | 6.25 |
| Matched Prompt (by GPT-5) | 0.969 | 0.948 | 0.941 | 31.59 | 5.93 |

- **(1) *Null Prompt*:** The optimization is driven solely by image and mask inputs, removing the text guidance entirely;

- **(2) *Incorrect Prompt*:** The model is provided with intentionally misinterpreted prompts (e.g., describing a rotation task as deformation) to simulate worst-case MLLM failure;

- **(3) *Matched Prompt*:** The standard configuration using prompts generated by MLLMs (comparing both GPT-5 and QWen-VL) based on the image and operation inputs.

The averaged quantitative results are presented in Tab. 5. As evidenced by the data, the framework remains robust even without textual guidance (i.e., *Null Prompt*), demonstrating the efficacy of the core region-level affine supervision.

In contrast, including *Matched Prompts* provides a valuable performance boost, whereas *Incorrect Prompts* degrade results due to conflicting guidance. However, thanks to the "Double Confirmation" workflow (detailed in Appendix D.3) we implemented, it helps prevent such misinterpretations in practice: rather than autonomous execution, the system requires users to select the optimal intent from generated candidates *before* optimization, ensuring the process is driven strictly by accurate, user-validated instructions.

## E    CRITERION DETAILS

To systematically evaluate the effectiveness of dragging operations, we introduce a set of quantitative criteria that assess both visual fidelity and spatial accuracy. These criteria fall into two complementary families: *Image Fidelity (IF)*, which measures the extent to which semantic content, source regions, and background integrity are preserved or altered as intended (detailed in Appendix E.1); and *Mean Distance (MD)*, which provides geometric and feature-based assessments of the dragging process (see Appendix E.2). Together, these metrics capture different yet complementary aspects of editing quality, enabling a fair and comprehensive evaluation of diverse dragging approaches.

### E.1    COMPUTATION OF IMAGE FIDELITY (IF)

**$\text{IF}_{s2t}$ Evalutaion.**    The identity fidelity from the source to the target (i.e., $\text{IF}_{s2t}$) evaluates how well original source features are preserved in the target regions after moving, promoting semantic consistency in dragged content. A higher $\text{IF}_{b2t}$ signifies superior fidelity, indicating minimal perceptual loss during the feature transfer. For each region $i$, we first affine-transform the masked original image to align with the target configuration using the full interpolation parameter of $k = K$:

$$\boldsymbol{x}_{\text{aff}} = \Omega \left( \boldsymbol{M}_i^{(0)} \odot \boldsymbol{x}, \ \ \boldsymbol{\xi}_i^{(K)} \right), \tag{28}$$

followed by masking both sides with the target region mask $\boldsymbol{M}_i^{(K)}$. Then the score is averaged as:

$$\text{IF}_{s2t} = 1 - \frac{1}{N} \sum_{i=1}^{N} LPIPS \left( \boldsymbol{M}_i^{(K)} \odot \boldsymbol{x}_{\text{aff}}, \ \ \boldsymbol{M}_i^{(K)} \odot \boldsymbol{x}' \right). \tag{29}$$

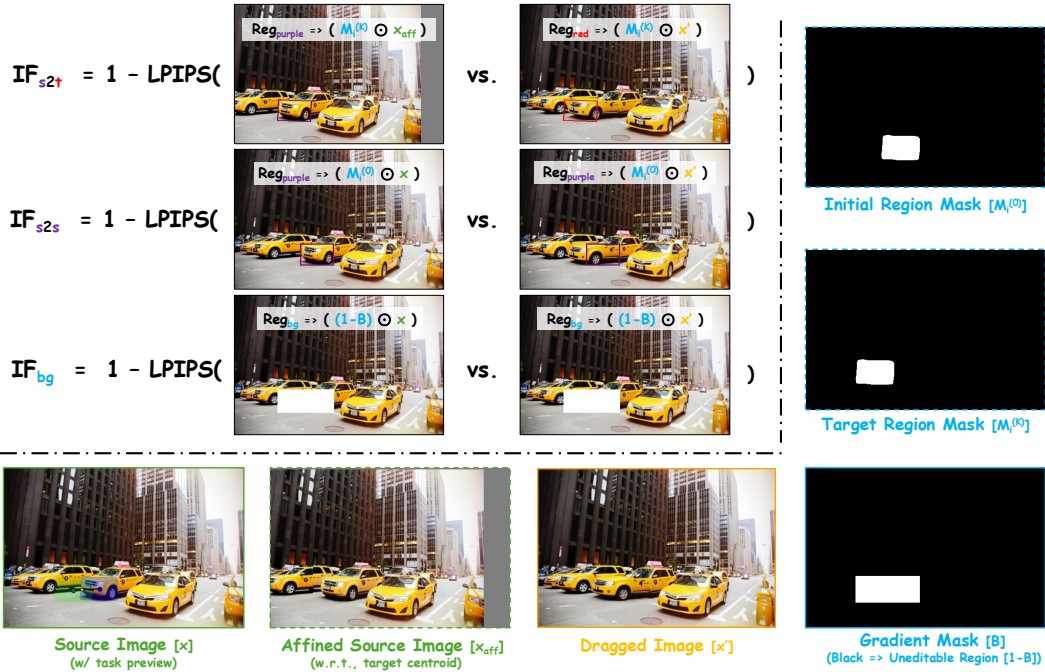

Figure 11: For $\text{IF}_{s2t}$ and $\text{IF}_{s2s}$, the criterion computation considers only the feature discrepancies within the labeled blocks. In $\text{IF}_{s2t}$, the purple region on the left image denotes $(M_i^{(K)} \odot x_{\text{aff}})$, corresponding to the source region of $(M_i^{(0)} \odot x)$, while the red patch on the right image represents the post-drag target $(M_i^{(K)} \odot x')$. By contrast, the criterion $\text{IF}_{s2s}$ compares the same purple original region $M_i^{(0)}$ across two images: the source $x$ (left) and the dragged result $x'$ (right). Lastly, $\text{IF}_{bg}$ evaluates all uneditable areas, as indicated by the black areas on the gradient mask as $(1 - B)$.

**$\text{IF}_{s2s}$ Evalutaion.** To assess the extent to which original features are effectively removed from the source regions after editing, we define the identity fidelity from source region to source region (i.e., $\text{IF}_{s2s}$), which measures the perceptual dissimilarity between the source region features in the original image $x$ and the edited image $x'$. A lower $\text{IF}_{s2s}$ indicates better performance, as it reflects greater divergence and implies successful "moving-out" of the selected features. Formally, for each source region $i$, we compute the *LPIPS* distance on region-masked image tensors and get the final mean score over all task regions via

$$\text{IF}_{s2s} = 1 - \frac{1}{N} \sum_{i=1}^{N} LPIPS\left(M_i^{(0)} \odot x, \ M_i^{(0)} \odot x'\right), \tag{30}$$

where $\odot$ indicates the element-wise multiplication, and $M_i^{(0)}$ indicates the original region mask (i.e., the operation region given by the user);

**$\text{IF}_{bg}$ Evalutaion.** To ensure background preservation outside edited regions, we introduce the background identity fidelity $\text{IF}_{bg}$, quantifying feature consistency in protected areas defined by the complement of the adaptive gradient mask $B$. A higher $\text{IF}_{bg}$ denotes better integrity, with minimal changes to non-targeted zones. Using the protection mask $(1 - B)$, we yield the score:

$$\text{IF}_{bg} = 1 - LPIPS\left((1 - B) \odot x, \ (1 - B) \odot x'\right). \tag{31}$$

To aid interpretation, Fig. 11 presents a visual example demonstrating how the three proposed criteria are applied in practice.

### E.2 COMPUTATION OF MEAN DISTANCE (MD)

**$\text{MD}_1$ Implementation.** We implement $\text{MD}_1$ following the existing criteria established by Xia et al. (2025). $\text{MD}_1$ enables precise feature matching within the uneditable region, allowing us to

validate the effectiveness of the dragging procedure by measuring the distance between the centroid of the source feature region and its most similar corresponding feature.

**$MD_2$ Implementation.** Building upon the original $MD_2$ design proposed in Lu et al. (2024a), we introduce an enhanced version that provides more precise and informative feedback for region-based drag operations. Unlike the original method, which computes feature matching distances based on manually annotated sample points, our approach automatically evaluates feature differences around the centroid scope of the pre- and post-drag regions. By leveraging this centroid-based formulation, we eliminate the inaccuracies and subjective biases inherent in manual annotations and also ensure a more consistent metric for assessing the effectiveness of various dragging strategies. This improvement allows for a more faithful reflection of the actual feature transformations induced by the dragging process and facilitates fairer comparisons across different methods.

## F    ADDITIONAL BASELINE INFORMATION

Here we provide the official project pages for the baseline methods used in our comparisons. All implementations follow the default configurations and instructions provided by their authors:

1 **CLIPDrag**: https://github.com/HKUST-LongGroup/CLIPDrag

2 **DragDiffusion**: https://github.com/Yujun-Shi/DragDiffusion

3 **DragLoRA**: https://github.com/Sylvie-X/DragLoRA

4 **DragNoise**: https://github.com/haofengl/DragNoise

5 **FastDrag**: https://github.com/XuanjiaZ/FastDrag

6 **FreeDrag**: https://github.com/LPengYang/FreeDrag

7 **GoodDrag**: https://github.com/zewei-Zhang/GoodDrag

8 **InstantDrag**: https://github.com/SNU-VGILab/InstantDrag

9 **RegionDrag**: https://github.com/Visual-AI/RegionDrag

And some encapsulated modules applied in our framework or experiments:

1 **InstantCharacter**: https://github.com/Tencent-Hunyuan/InstantCharacter

2 **SD3.5-Large-IP-Adapter**: https://huggingface.co/InstantX/SD3.5-Large-IP-Adapter

## G    BENCHMARK DETAILS

### G.1    FORMATION OF THE *ReD* BENCHMARK

To evaluate model performance on the regional drag-based image editing task, we introduce a new benchmark, the *Regional-based Dragging (ReD)* Bench, consisting of 120 images annotated with precise drag instructions at both point and region levels. Each manipulation in the dataset is associated with an intention label, selected from *relocation*, *deformation*, or *rotation*.

For every image, we provide two complementary instruction sets corresponding to point-based and region-based dragging. The region-based annotations are supplied as multiple PNG masks, with each region uniquely represented by its centroid for cross-reference. The drag annotations include multiple start-to-target point pairs, which can be directly aligned with the region annotations, ensuring consistency in task intention. Additionally, we provide background prompts and editing intention prompts for each image to facilitate multimodal tasks, along with masks generated using the DragFlow automatic masker Appendix D.2. These design choices enable a more faithful representation of user intents underlying the provided drag instructions.

### G.2    ADOPTION OF THE *DragBench-DR* BENCHMARK

To further assess the effectiveness of DragFlow on a broader spectrum of images, we adapt and evaluate it on *DragBench-DR* (Lu et al., 2024a). *DragBench-DR* extends the classic point-based

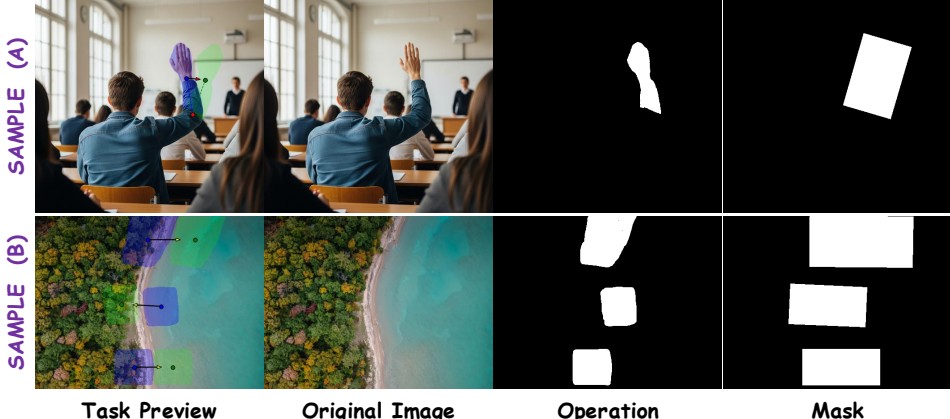

Figure 12: Real samples from *ReD* Bench: the **first colume** shows the dragging preview, where the green region is estimated from the user-specified target centroid; the **second colume** presents the source images, while the **third colume** highlights the user-marked operation regions in the form of masks, which may include multiple valid regions; and the **last colume** depicts the adaptive masks generated (detailed in Appendix D.2). Other matched instructions are provided in Appendix G.3.

dragging benchmark *DragBench* to region-based operations. Unlike the original benchmark, which relies on sparse point guidance, *DragBench-DR* formulates edits over regions, thereby providing a clearer reflection of user intentions. For evaluation, the accompanying metrics compare the pre-drag source region with the post-drag target region by computing differences over pre-annotated correspondence points. Despite this extension, as noted by the authors, *DragBench-DR* remains consistent with its point-based counterpart, while more effectively capturing region-level semantics in interactive editing tasks.

While *DragBench-DR* extends the benchmark to region-based operations, its evaluation protocol ($MD_2$) still relies on point comparisons: differences are computed between pre- and post-drag regions using pre-annotated correspondence points. This design can introduce mismatches, as region-level edits are not faithfully captured by sparse point feature correspondences, leading to potentially unfair assessments for region methods. To better align the evaluation with region-based editing and integrate existing datasets into our experiment, we update the feature comparison criterion by replacing point-based annotations with an automatic centroid-based formulation (see Appendix E.2).

### G.3 DEMONSTRATION OF DATA SAMPLES

We present two real data samples (i.e., SAMPLE (A) and SAMPLE (B)) from the *ReD* Bench. The corresponding instructions are provided as follows, and the images are shown in Fig. 12.

```
{ % SAMPLE (A) %
    "region_operations": {
        "0": {
            "task": "rotation",
            "centroids": [[337, 175], [379, 179]],
            "anchors": [351, 256]
        }
    },
    "point_operations": {
        "begin_points": [[326, 111], [342, 190]],
        "target_points": [[400, 116],[376, 198]]
    },
    "background_prompt": "From a rear view, a student in a blue denim
        jacket raises their hand in a classroom. Wooden desks, large
        windows (letting in light), and a distant teacher form the
        backdrop. The scene captures an engaged learning moment, with a
        realistic, observational style.",
    "editing_prompt": " The student in a blue denim jacket moves his arm
        rightward, with his hand closer to the right side on this image."
}
```

```
1  { % SAMPLE (B) %
2      "region_operations": {
3          "0": {
4              "task": "deformation",
5              "centroids": [[251, 52], [357, 52]],
6              "anchors": null
7          },
8          "1": {
9              "task": "deformation",
10             "centroids": [[281, 200], [192, 195]],
11             "anchors": null
12         },
13         "2": {
14             "task": "deformation",
15             "centroids": [[221, 335], [307, 335]],
16             "anchors": null
17         }
18     },
19     "point_operations": {
20         "begin_points": [[284, 11], [244, 96], [280, 165], [287, 235],
                   [243, 305], [244, 365]],
21         "target_points": [[392, 11], [356, 97], [193, 162], [199, 233],
                   [332, 306], [335, 367]]
22     }
23     "background_prompt": "The image is an aerial view of a coastal scene.
              There's a beach with light - colored sand between a dense forest
              (with green, yellow, and orange foliage) and a turquoise - blue
              sea. The forest covers the left side, the beach runs along the
              middle, and the sea is on the right.",
24     "editing_prompt": "The top and bottom sections of the beach are
              narrowed to the outside, and the middle part is narrowed inside,
              altering the coastline shape to form a bay."
25 }
```

## H EXTRA QUALITATIVE RESULTS

### H.1 ADDITIONAL QUALITATIVE SAMPLES

In addition to the qualitative studies reported in the main experiments, we provide further examples in Fig. 15 and Fig. 16. These additional visualizations help to illustrate the advantages of our approach across diverse editing scenarios.

### H.2 CHALLENGING SCENARIOS

To further assess the robustness of DragFlow, we evaluate its performance under extreme conditions. As illustrated in Fig. 14, our analysis focuses on two distinct dimensions: **(a) complex operational requirements** and **(b) unusual feature structures**. Following this, in Subsec. H.3, we are also advised to provide some illustrations about the failure cases. We believe these parts may inspire and offer insights for future research about image drag-editing.

**(a) Cases with Complex Instructions.** Fundamentally, we conceptualize the defined operations as elementary "building blocks." Consequently, sophisticated instructions that involve composite movements (e.g., do both rotation and relocation) or non-affine warps (e.g., bending) are effectively executed by composing these basic units. Among the composite movements, we consider **two** scenarios, where multiple distinct operations are processed in parallel or in sequence. The lower section of Fig. 14 demonstrates how our framework effectively handles these complex instructions and overcomes the challenging scenarios.

**(b) Cases with Complex Structures.** Regarding the non-rigid deformations inherent to complex textures such as hair and cloth, our empirical results confirm that the affine assumption remains valid.

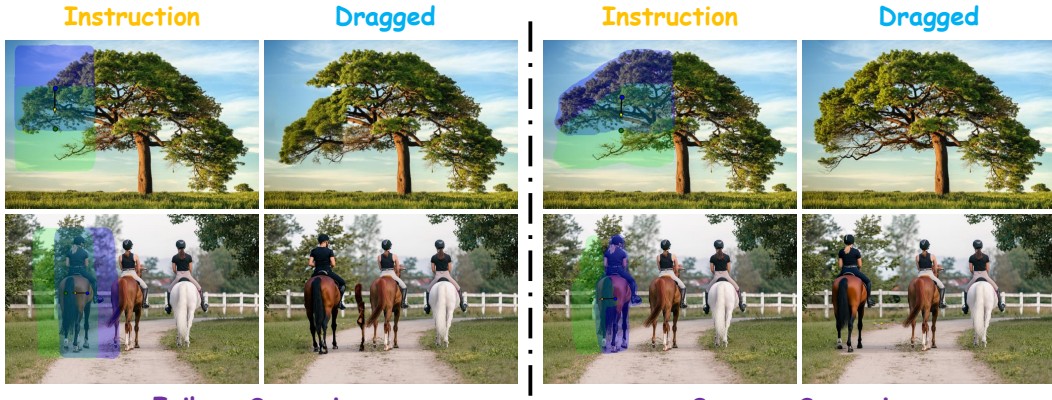

Figure 13: Extra qualitative results for failure scenarios. **Left**: The incorrect operation mask and failure outcomes; **Right**: The suitable operation mask with failure outcomes.

By applying flexible single operations or iteratively composing multiple blocks to approximate the target motion, the framework maintains high editing effectiveness and structural fidelity even on these intricate features. Representative samples validating this capability are presented in the upper part of Fig. 14.

### H.3    FAILURE SCENARIOS

**Dependency on Input Precision.**    A primary limitation of the proposed framework is its reliance on the quality of user-provided inputs. Given that both the region-level affine supervision and the adaptive gradient masking mechanisms hinge directly on the initial mask annotation, ambiguous or inaccurate inputs can inevitably introduce generation artifacts.

Empirically, we observe that while the framework exhibits considerable tolerance for slightly over-sized masks, ensuring complete coverage of the targeted semantic object is critical. Specifically, undersized or incomplete masks tend to compromise the editing quality. Fig. 13 illustrates this phenomenon by comparing outcomes derived from precise versus careless annotations. To relax these strict precision requirements, future iterations could explore integrating auxiliary techniques such as superpixel segmentation or implementing error-tolerant peripheral buffer zones to robustly handle manual variances.

## I    RUNTIME AND MEMORY COMPARISON

To provide a comprehensive understanding of DragFlow's practical performance, especially as an optimization-based method, a quantitative analysis of its computational cost is necessary. This discussion is intended to help users understand the quality versus speed trade-offs inherent in the approach. Tab. 6 presents a detailed comparison of the average inference time and GPU memory consumption for DragFlow alongside various baseline methods. The results were recorded as an average over all samples contained in *DragBench-DR* using one NVIDIA H100 GPU.

In our implementation, we applied qint8 quantization to the FLUX backbone to accelerate inference and reduce memory usage during the optimization. Additionally, by enabling *CPU Offloading*, the framework is capable of running on a consumer-grade 24 GB GPU. Beyond the current level, we believe there is still plenty of room to further reduce memory costs and increase speed through future improvements in both the algorithm and the code.

As the data illustrates, DragFlow's resource requirements are higher than the baseline methods. This is an anticipated result and a direct consequence of the underlying model architecture. DragFlow is the only method in this comparison engineered to operate on the FLUX.1 model, a high-parameter, DiT-based architecture. In contrast, all other baselines are built upon the significantly smaller and less complex Stable Diffusion models.

Table 6: This table collects and summarizes the inference time and memory consumption depending on different image drag-editing methods. Among them, DragFlow, as the only DiT-based dragger, shows the highest resource demand due to its more complex backbone architecture. The results are recorded as averages on *DragBench-DR*.

| Methods | Prep + Edit Time (s) | Peak Memory (GB) |
|---|---|---|
| RegionDrag | 9.2 | 4.3 |
| FastDrag | 5.4 | 5.2 |
| InstantDrag | 1.2 | 4.1 |
| DragLoRA | 58.6 | 14.6 |
| FreeDrag | 113.2 | 13.2 |
| DragNoise | 103.4 | 12.2 |
| GoodDrag | 104.6 | 12.8 |
| CLIPDrag | 101.0 | 19.4 |
| DragDiffusion | 82.1 | 12.1 |
| **DragFlow** | 158.7 | 23.5 |

Table 7: The editing performance of our DragFlow framework on different DiT-based generative priors (SD3.5 vs. FLUX.1). The results are averaged on *DragBench-DR*.

| Method | Image Fidelity | | | Mean Distance | |
|---|---|---|---|---|---|
| | $\mathbf{IF}_{bg} \uparrow$ | $\mathbf{IF}_{s2t} \uparrow$ | $\mathbf{IF}_{s2s} \downarrow$ | $\mathbf{MD}_1 \downarrow$ | $\mathbf{MD}_2 \downarrow$ |
| DragFlow (based on Stable Diffusion 3.5) | 0.962 | 0.945 | 0.952 | 35.21 | 6.58 |
| DragFlow (based on FLUX.1-dev) | 0.969 | 0.948 | 0.941 | 31.59 | 5.93 |

## J  FRAMEWORK AND MODULE GENERALIZABILITY

### J.1  GENERALIZABILITY OF DRAGGING FRAMEWORK

To validate the generalization capability of DragFlow beyond the specific architecture of FLUX, we conducted an additional evaluation applying our framework to *Stable Diffusion 3.5* (SD3.5), another prominent text-to-image model based on the DiT architecture. Notably, since the default *InstantCharacter* adapter is FLUX-specific, we employed *SD3.5-Large-IP-Adapter* to ensure a fair comparison. Tab. 7 presents the comparative editing performance averaged over all samples from the *DragBench-DR* dataset. As the results indicate, DragFlow maintains robust performance when transferred to the SD3.5 backbone.

Specifically, the SD3.5-based implementation achieves an $\mathbf{MD}_1$ of 35.21 and an $\mathbf{IF}_{s2t}$ of 0.945, versus 31.59 and 0.948 of the FLUX edition, respectively. The FLUX-based version demonstrates a slight performance advantage, likely due to its substantially larger parameter count and enhanced generative prior, while the performance on SD3.5 remains competitive and consistent. These findings provide empirical evidence that the key contributions of DragFlow, including region-level affine supervision, gradient mask constraints, and adapter-enhanced inversion, are agnostic to the architecture used. The framework effectively generalizes across different DiT-based backbones, confirming that its efficacy is not limited to the specific training paradigm of FLUX but rather serves as a generalized solution for modern DiT generative models.

### J.2  GENERALIZABILITY OF ID-PRESERVATION DESIGN

To further evaluate the design choices behind our method, we conducted an experiment in which the IP-Adapter was replaced with a subject-specific LoRA. In this setup, a LoRA was first trained for the target subject and then applied during the drag-editing stage. Performance on DragBench-DR is summarized in Table 8.

The LoRA-based variant exhibits slightly stronger performance, which aligns with expectations since the LoRA is expressly optimized for the specific subject rather than functioning as a generic,

Table 8: The editing performance of our DragFlow framework on different ID-preservation modules (IP-Adapter vs. LoRA). The results are averaged on *DragBench-DR*.

| Method | Image Fidelity | | | Mean Distance | |
|---|---|---|---|---|---|
| | $\mathbf{IF}_{bg} \uparrow$ | $\mathbf{IF}_{s2t} \uparrow$ | $\mathbf{IF}_{s2s} \downarrow$ | $\mathbf{MD}_1 \downarrow$ | $\mathbf{MD}_2 \downarrow$ |
| DragFlow (w/ LoRA) | 0.970 | 0.949 | 0.940 | 30.98 | 5.88 |
| DragFlow (w/ IP-Adapter) | 0.969 | 0.948 | 0.941 | 31.59 | 5.93 |

open-domain adapter. However, this improvement comes with the overhead of training a new LoRA for each subject. In contrast, many modern text-to-image models now include high-quality, readily usable IP-Adapters, making the IP-Adapter approach more practical and broadly applicable in typical workflows.

## K  LLM USAGE STATEMENT

We used large language models for text polishing and grammar correction during manuscript preparation. No LLMs were involved in the design of the method, experiments, or analysis. All content has been carefully verified and validated by the authors.

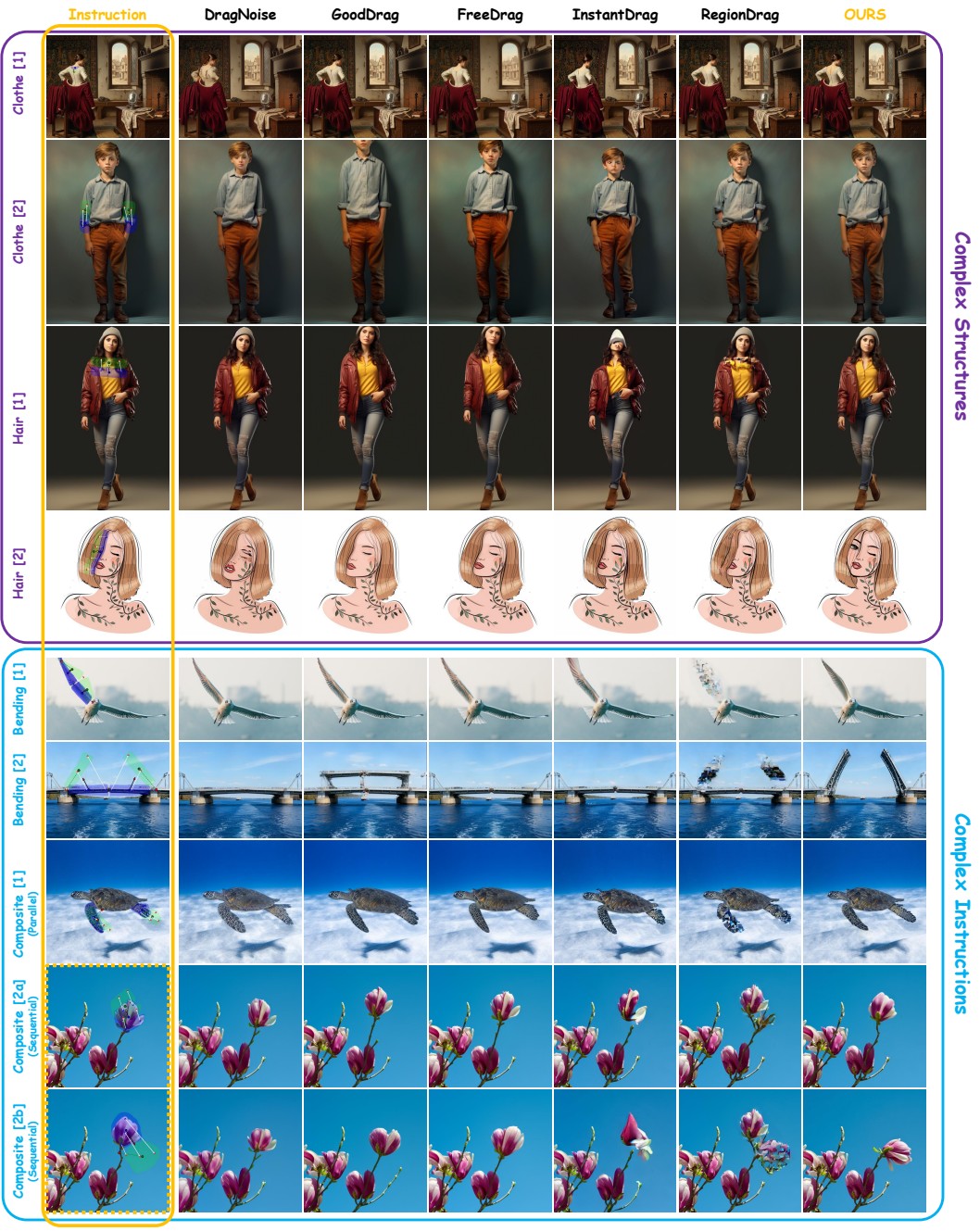

Figure 14: Extra qualitative comparison for challenging scenarios, with **complex feature structures** (e.g., clothes and hair) or **complex operational instructions** (e.g., bending and composite operations). Two cases of composite operations are provided: the parallel and the sequential.

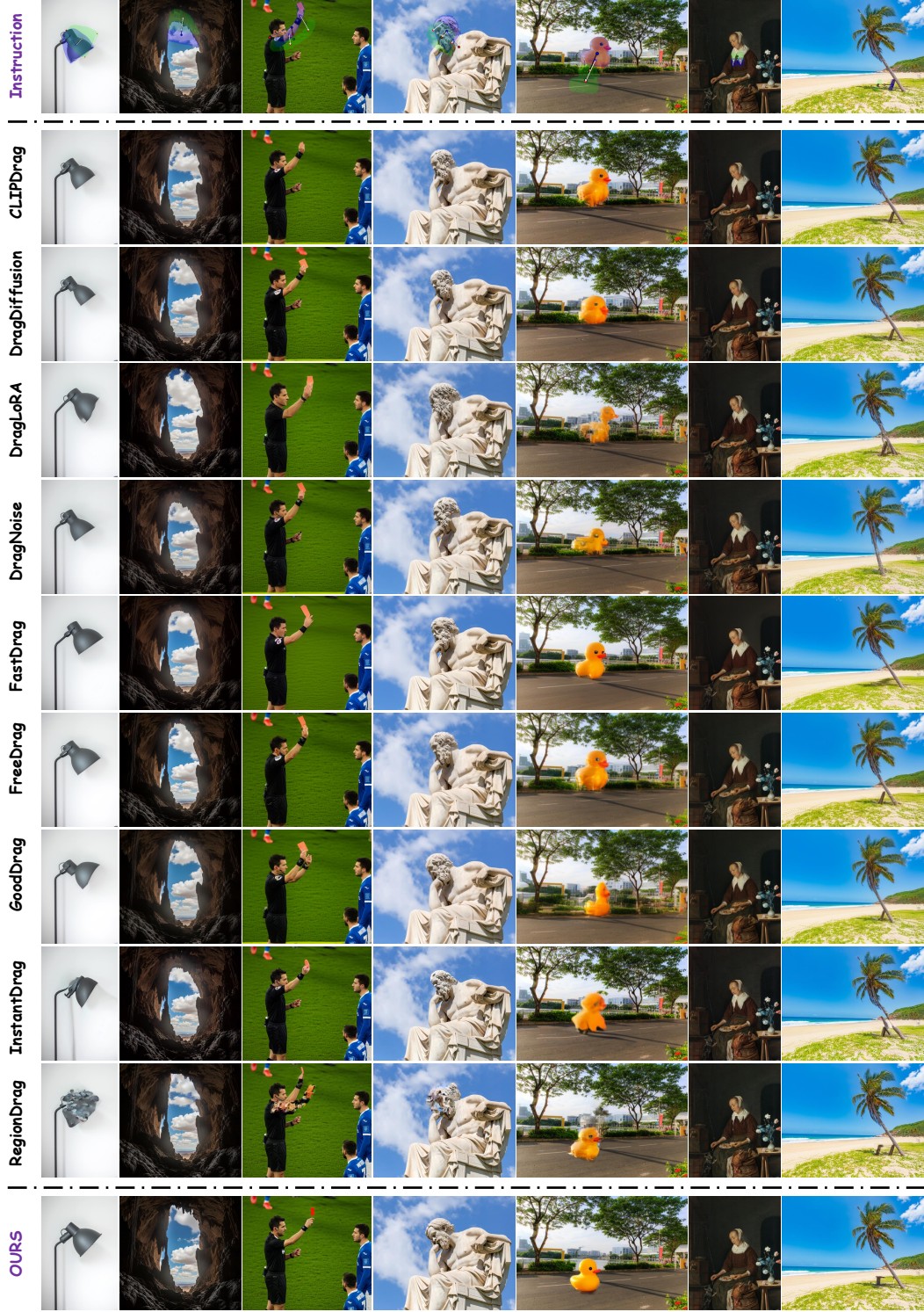

Figure 15: Extra qualitative comparison (Part 1 out of 2) of DragFlow with multiple baselines.

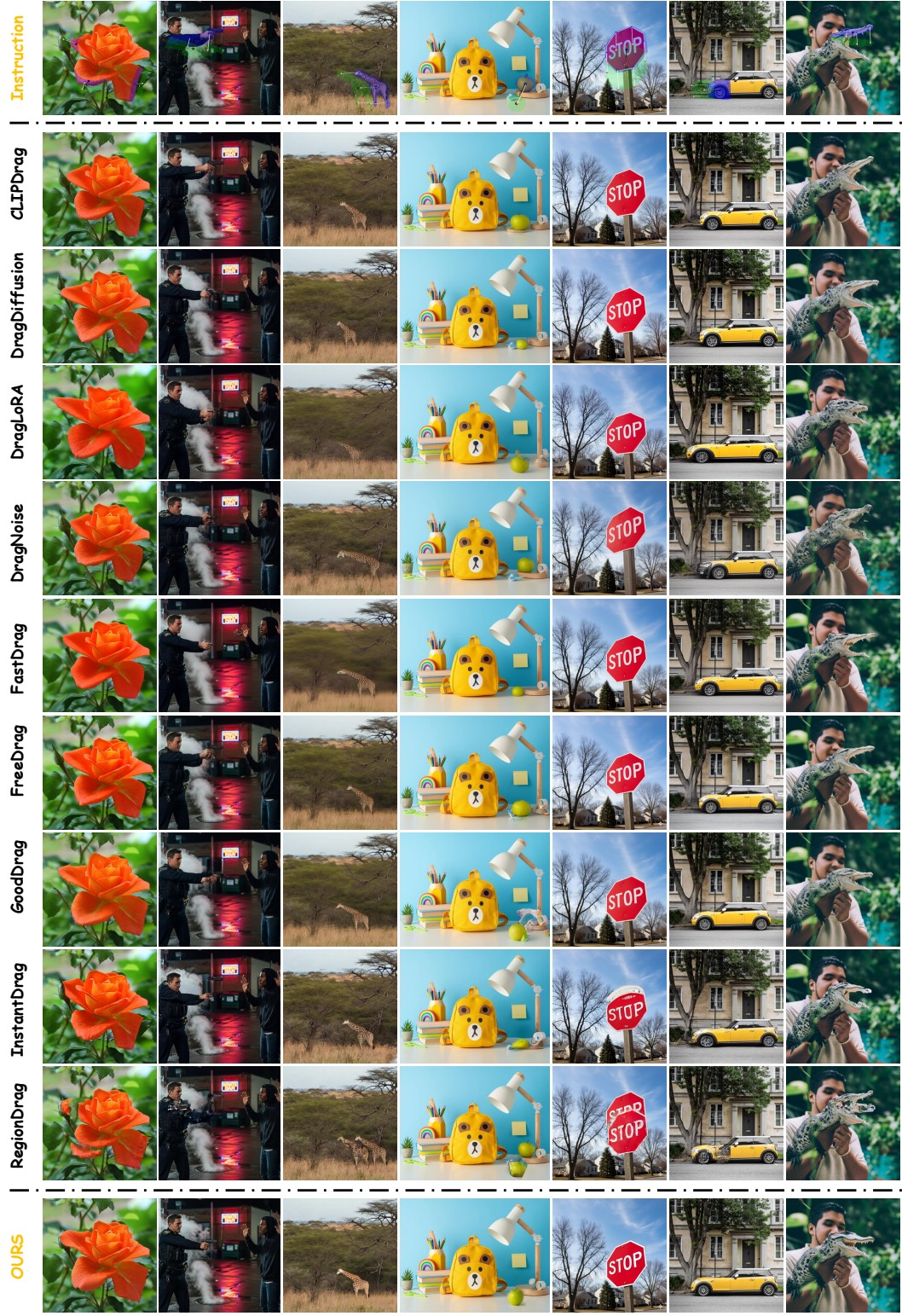

Figure 16: Extra qualitative comparison (Part 2 out of 2) of DragFlow with multiple baselines.

