# OpenReview forum: "DragFlow: Unleashing DiT Priors with Region-Based Supervision for Drag Editing"
_ICLR.cc/2026/Conference — ICLR 2026 Poster_

### Official Review · Reviewer_ZsfJ · 2025-10-27

**Soundness:** 3
**Presentation:** 2
**Contribution:** 3
**Rating:** 6
**Confidence:** 3

**Summary:**

This paper proposes a region-based framework for drag-based image editing using Diffusion Transformers (DiTs) instead of UNet-based diffusion models. The authors argue that point-based supervision fails on DiTs due to overly fine-grained features and address this with region-level learning. They also introduce a new benchmark extending DragBench with region annotations, achieving state-of-the-art results as shown in Table 2, with further component analysis in Table 3.

**Strengths:**

1: The paper provides a meaningful analysis of why DiT-based models fail with point-based drag supervision. This motivation is well grounded and timely, as future generative systems are likely to rely on DiTs rather than UNet backbones.

2: The method achieves clear state-of-the-art performance on both ReD Bench and DragBench-DR, showing consistent improvements over existing drag-based editing approaches.

**Weaknesses:**

1: The paper does not provide any discussion or quantitative analysis of runtime or inference time. Reporting the average optimization time per image or per drag operation would make the comparison with prior work more complete.

2: The paper mainly presents successful examples but does not include qualitative results in more challenging scenarios such as complex backgrounds or strong occlusions. Showing a few representative failure cases or difficult examples would help readers better understand the method’s limitations and potential failure modes.

3: Section 3.4 mainly integrates existing personalization adapters to improve inversion, which is effective but not novel. More discussion on how the adapter interacts with DiT features would clarify the contribution.

Minor suggestion:
1 The notion of a “stronger generative prior” is repeatedly emphasized in the abstract and introduction, yet its meaning and mechanism of utilization in DiT remain unclear. The underlying reasoning of why and how this prior facilitates more faithful or controllable edits only becomes explicit in Section 3.1–3.2. It would be helpful to briefly clarify in the abstract how this stronger prior is actually leveraged/achieved by your framework, perhaps in one sentence.

2 The analysis in Section 3.1 relies mainly on qualitative visualization to argue that point-based supervision fails on DiT due to finer feature granularity. While reasonable, the claim lacks quantitative evidence, such as metrics of using DiT to support the argument.

3 It is unclear which dataset was used for the ablation in Table 3.

**Questions:**

1: The method assumes that all drag operations can be represented by affine transformations such as translation, deformation, or rotation. How well does this assumption hold for complex non-rigid deformations, for example hair, cloth, or water surfaces?

2: Which layer of the DiT the feature extractor F(⋅) is taken for region-level supervision. Is the performance sensitive to this choice, and has any analysis been conducted on different feature layers?

---

> ### Author Response · Authors · 2025-11-24
> **Response to Reviewer ZsfJ (1/2)**
>
> Dear Reviewer ZsfJ,
>
> Thank you for dedicating time to this insightful review. Your constructive comments have been instrumental in refining our work.
>
> We are particularly encouraged by your recognition that our analysis of point-based supervision on DiTs is "meaningful" and "timely," providing a well-grounded motivation. We also sincerely value your validation of our method's "clear state-of-the-art performance" and "consistent improvements" across benchmarks.
>
> We value all your feedback and will address your concerns piece by piece.
>
> ---
>
> **[W1] The paper does not provide any discussion or quantitative analysis of runtime or inference time. Reporting the average optimization time per image or per drag operation would make the comparison with prior work more complete.**
>
> We sincerely thank you for reminding this. The comparison of wall-clock runtime and memory is summarized in the table below. The results are averaged on *DragBench-DR* using NVIDIA H100 GPU. We have included this information in ***Appendix I (Skyblue)***.
>
> |Methods|Prep + Edit Time (s)|Peak Memory (GB)|
> |-|-|-|
> |RegionDrag|9.2|4.3|
> |FastDrag|5.4|5.2|
> |InstantDrag|1.2|4.1|
> |DragLoRA|58.6|14.6|
> |FreeDrag|113.2|13.2|
> |DragNoise|103.4|12.2|
> |GoodDrag|104.6|12.8|
> |CLIPDrag|101.0|19.4|
> |DragDiffusion|82.1|12.1|
> |DragFlow|158.7|23.5|
>
> As the data present, DragFlow is **slower and more memory-consuming than methods based on much smaller SD models**. This is an expected result, as DragFlow is **the first one designed to leverage the significantly larger and more powerful DiT base models**.
>
> In our implementation, we use **qint8 quantization** to accelerate inference and reduce memory usage, and pair it with **CPU offloading** to further decrease GPU memory requirements. Beyond the current level, we believe there is still plenty of room to further reduce memory costs and increase speed through future improvements in both the algorithm and the code.
>
> ---
>
> **[W2] The paper mainly presents successful examples but does not include qualitative results in more challenging scenarios such as complex backgrounds or strong occlusions. Showing a few representative failure cases or difficult examples would help readers better understand the method’s limitations and potential failure modes.**
>
> Thank you very much for your valuable suggestion. We have now included several representative failure cases in ***Figure 13*** to better illustrate the limitations of our framework.
>
> A key limitation of our method is its **reliance on reasonably accurate user-provided scribbles**. The scribbles do not need to precisely trace the object boundary, and a slightly larger region around the object usually **does not affect** the results. However, if the scribbles include large portions of the background, noticeable artifacts may appear.
>
> ***Figure 13*** shows two failure cases. Due to imprecise masking, unrelated background regions are also dragged (the sky in the first row and a portion of the horse in the second), leading to unintended results.
>
> We sincerely appreciate your feedback, which has helped us further improve the completeness of our work.
>
> ---
>
> **[W3] Section 3.4 mainly integrates existing personalization adapters to improve inversion, which is effective but not novel. More discussion on how the adapter interacts with DiT features would clarify the contribution.**
>
> Thank you for this constructive comment. While personalization adapters are indeed established tools, our contribution lies in being **the first to employ them to alleviate the “inversion drift” problem inherent to CFG-distilled models**, thereby improving drag-based editing.
>
> Regarding the interaction between the IP adapter and the DiT features: the IP adapter extracts more informative subject-specific hidden states and adds them to the original hidden states in Flux. This **rectifies the latent trajectory**, helping the optimized latents **remain faithful to the subject’s identity**.
>
> ---
>
> **[Minor Suggestion 1] It would be helpful to briefly clarify in the abstract how this stronger prior is actually leveraged/achieved by your framework, perhaps in one sentence.**
>
> Thank you for your valuable suggestion. Following your advice, we have revised the abstract to briefly clarify how this stronger prior is actually leveraged.

---

> ### Author Response · Authors · 2025-11-24
> **Response to Reviewer ZsfJ (2/2)**
>
> **[Minor Suggestion 2] The analysis in Section 3.1 relies mainly on qualitative visualization to argue that point-based supervision fails on DiT due to finer feature granularity. While reasonable, the claim lacks quantitative evidence, such as metrics of using DiT to support the argument.**
>
> We appreciate the reviewer’s insightful comment and agree that quantitative evidence is essential to strengthen the analysis in ***Section 3.1***.
>
> To address this, we conducted an additional experiment comparing the point-based and region-based variants on *DragBench-DR*, with both methods equipped with **Background Preservation** and **Adapter-Enhanced Inversion** for a fair comparison. The results are summarized in the table below:
>
> | **Attempts**                  | **$\text{IF}_\text{bg}$ ↑** | **$\text{IF}_\text{s2t}$ ↑** | **$\text{IF}_\text{s2s}$ ↓** | **$\text{MD}_1$ ↓** | **$\text{MD}_2$ ↓** |
> |------------------------------|------------------------------|-------------------------------|-------------------------------|----------------------|----------------------|
> | Point-based variant          | 0.961                        | 0.943                         | 0.958                         | 37.02                | 6.94                 |
> | Region-based variant (DragFlow) | 0.969                        | 0.948                         | 0.941                         | 31.59                | 5.93                 |
>
> The substantial improvements in $\text{MD}_1$ and $\text{MD}_2$ clearly indicate that **sparse point-based supervision is less effective for DiT architectures**. These quantitative findings support our observation in ***Section 3.1*** that DiT’s fine-grained feature representations **lack the semantic coherence** needed for reliable point tracking, whereas region-based supervision effectively compensates for this limitation and better unleash the model's potential.
>
> ---
>
> **[Minor Suggestion 3]  It is unclear which dataset was used for the ablation in Table 3.**
>
> We apologize for this earlier omission. We have revised the caption of ***Table 3*** to explicitly state that our ablation studies were conducted on the *ReD Bench*.
>
> ---
>
> **[Q1] The method assumes that all drag operations can be represented by affine transformations such as translation, deformation, or rotation. How well does this assumption hold for complex non-rigid deformations, for example hair, cloth, or water surfaces?**
>
> Thank you very much for this valuable question. We view the three defined operations as fundamental **building blocks**. By using these building blocks, either individually or in combination, we can effectively **approximate a wide range of complex non-rigid deformations**, including challenging cases such as hair and cloth.
>
> To further illustrate this, we have added representative results in ***Figure 14***, which include examples involving non-rigid structures (for example, cloth and hair). These results show that our approach maintains editing fidelity in these more complex scenarios.
>
> We sincerely appreciate your suggestion and would be happy to clarify or expand on any additional points.
>
> ---
>
> **[Q2] Which layer of the DiT the feature extractor F(⋅) is taken for region-level supervision. Is the performance sensitive to this choice, and has any analysis been conducted on different feature layers?**
>
> Thank you for this thoughtful question. For region-level supervision, we use the features from the **17th and 18th double-stream blocks** of the DiT. This choice was initially guided by a small-scale ablation study conducted in the early stages of our work.
>
> To more thoroughly assess the sensitivity of our results to this design choice, we carried out an extensive ablation study using features from all layers. The key metrics are shown in ***Figure 9***.
>
> As shown in the results, the **features from most double-stream blocks outperform** those from the single-stream blocks, with the **17th and 18th blocks achieving the best** overall performance. We sincerely appreciate your feedback, which has helped us further strengthen the completeness and clarity of our work.

---

### Official Review · Reviewer_yXpA · 2025-10-30

**Soundness:** 3
**Presentation:** 3
**Contribution:** 3
**Rating:** 8
**Confidence:** 3

**Summary:**

This paper presents DragFlow, a drag-based editing method specifically designed for Diffusion Transformers (DiTs). The authors' core insight is that point-based supervision, standard in prior work, is fundamentally mismatched with the fine-grained feature structure of DiTs. They solve this by introducing a region-level supervision scheme that treats the object as a whole, guiding the generation process with progressive affine transformations. This is combined with a clever adapter-based inversion technique to preserve subject identity and a hard-constrained background mask. The method sets a new state-of-the-art, producing more realistic and structurally coherent results on a new, more challenging benchmark (ReD Bench) also introduced by the authors.

**Strengths:**

1.  **Clear, Fundamental Insight:** The paper's main strength is its correct diagnosis of *why* prior methods fail on DiTs. The shift from point- to region-level supervision isn't just an incremental tweak; it's a necessary conceptual shift grounded in a solid understanding of the underlying model architectures. This is a valuable contribution.

2.  **Effective and Cohesive System:** The proposed solution is elegant and well-engineered. Region-level supervision provides robust guidance, while the adapter-enhanced inversion is a very practical solution to the known identity drift problem in distilled models like FLUX. The components work together to solve the problem comprehensively.

3.  **Strong Empirical Support:** The experiments are convincing. DragFlow clearly outperforms a wide range of competitors on both quantitative metrics and visual quality, especially in complex non-rigid deformations where other methods falter. The ablation study effectively validates the importance of each design choice, and the new ReD Bench is a welcome contribution for future research in this area.

**Weaknesses:**

1.  **Dependency on External Adapters:** The method's excellent identity preservation relies on a pre-trained IP-Adapter. This is a pragmatic choice, but it means performance is coupled to the quality of this external module. The paper would be stronger with a brief discussion on this dependency.

2.  **Missing Efficiency Analysis:** The method is iterative and involves multiple components. A quantitative comparison of the runtime against faster, non-optimizing methods (like FastDrag) is missing. This would help users understand the quality vs. speed trade-off. Surpassing non-optimizing methods in speed is not necessary, but a discussion section is necessary for user's information.\

3. Some prior works (like LightningDrag InstantDrag) predict dense motion field from user's sparse input, can their dense motion be used as region supervision in DragFlow?

**Questions:**

- The MLLM front-end for intent parsing is a nice usability feature. How robust is it in practice? What happens if it misinterprets the user's intended transformation (e.g., deformation vs. rotation), and is there a simple way for the user to correct it?

- The framework currently handles relocation, deformation, and rotation via affine transforms. How would it generalize to more complex, non-affine warps like twisting or bending?

---

> ### Author Response · Authors · 2025-11-24
> **Response to Reviewer yXpA (1/2)**
>
> Dear Reviewer yXpA,
>
> We truly value your constructive feedback. Your thoughtful comments have greatly contributed to enhancing our work.
>
> We are particularly encouraged by your recognition of our DiT feature analysis as a "fundamental insight" and a "necessary conceptual shift". We also deeply appreciate your validation of DragFlow as an "effective and cohesive system" with "strong empirical support", especially regarding complex deformations and the contribution of *ReD Bench*.
>
> In the following sections, we are glad to provide a detailed explanation and address your remaining questions point by point:
>
> ---
>
> **[W1] Dependency on External Adapters: The method's excellent identity preservation relies on a pre-trained IP-Adapter. This is a pragmatic choice, but it means performance is coupled to the quality of this external module. The paper would be stronger with a brief discussion on this dependency.**
>
> Thank you very much for this thoughtful and constructive comment. We fully understand your concern that the performance of our method may be influenced by the quality of the pre-trained IP-Adapter. Indeed, this dependency is an important practical consideration.
>
> To address this point, we conducted an additional experiment where we replaced the IP-Adapter with a subject-specific LoRA. In this setting, we first **trained a LoRA for the target subject** and then used it during the drag-editing process. The results on *DragBench-DR* are shown below.
>
> |**Method** | **$\text{IF}_\text{bg}$ ↑** | **$\text{IF}_\text{s2t}$ ↑** | **$\text{IF}_\text{s2s}$ ↓** | **$\text{MD}_\text{1}$ ↓** | **$\text{MD}_\text{2}$ ↓** |
> |----------------------------|-------|-------|-------|-------|------|
> |DragFlow (w/ LoRA)          | 0.970 | 0.949 | 0.940 | 30.98 | 5.88 |
> |**DragFlow (w/ IP-Adapter)**| 0.969 | 0.948 | 0.941 | 31.59 | 5.93 |
>
> Interestingly, the **LoRA-based variant performs slightly better**, which is expected because LoRA is tailored to the specific subject rather than serving as a generic, open-domain adapter. Meanwhile, this comes at the cost of additional training time. Nonetheless, **many modern text-to-image models have high-quality, compatible IP-Adapters**, making the default approach practical and widely applicable.
>
> We have incorporated these experiments into ***Appendix J.2 (Skyblue)***. If you have further suggestions, we would be glad to continue the discussion. Your feedback is genuinely helpful in strengthening our work.
>
> ---
>
> **[W2] Missing Efficiency Analysis: The method is iterative and involves multiple components. A quantitative comparison of the runtime against faster, non-optimizing methods (like FastDrag) is missing. This would help users understand the quality vs. speed trade-off. Surpassing non-optimizing methods in speed is not necessary, but a discussion section is necessary for user's information**
>
> We sincerely thank you for reminding this. The comparison of wall-clock runtime and memory is summarized in the table below. The results are averaged on *DragBench-DR* using NVIDIA H100 GPU. We have included this information in ***Appendix I (Skyblue)***.
>
> |Methods|Prep + Edit Time (s)|Peak Memory (GB)|
> |-|-|-|
> |RegionDrag|9.2|4.3|
> |FastDrag|5.4|5.2|
> |InstantDrag|1.2|4.1|
> |DragLoRA|58.6|14.6|
> |FreeDrag|113.2|13.2|
> |DragNoise|103.4|12.2|
> |GoodDrag|104.6|12.8|
> |CLIPDrag|101.0|19.4|
> |DragDiffusion|82.1|12.1|
> |DragFlow|158.7|23.5|
>
> As the data present, DragFlow is **slower than methods based on much smaller SD models**. This is an expected result, as DragFlow is **the first one designed to leverage the significantly larger and more powerful DiT base models**.
>
> In our implementation, we use **qint8 quantization** to accelerate inference and reduce memory usage, and pair it with **CPU offloading** to further decrease GPU memory requirements. Beyond the current level, we believe there is still plenty of room to further reduce memory costs and increase speed through future improvements in both the algorithm and the code.
>
> ---
>
> **[W3] Some prior works (like LightningDrag InstantDrag) predict dense motion field from user's sparse input, can their dense motion be used as region supervision in DragFlow?**
>
> Thank you for this insightful question. One potential way to incorporate dense motion fields is to **replace the affine transformation with a dense field**, allowing the source mask to be warped according to the per-pixel motion vectors at each optimization step.
>
> However, doing so would delegate the entire interpretation of the user’s editing intent **to an external module**. In our observations, current **dense-motion prediction models often fall short** compared to MLLMs in accurately translating user intent.
>
> We believe that if dense-motion extractors mature to the point where they can reliably resolve ambiguities, this would be a promising direction to explore. If you have additional thoughts or suggestions, we would be very happy to continue the discussion.

---

> ### Author Response · Authors · 2025-11-24
> **Response to Reviewer yXpA (2/2)**
>
> **[Q1] The MLLM front-end for intent parsing is a nice usability feature. How robust is it in practice? What happens if it misinterprets the user's intended transformation (e.g., deformation vs. rotation), and is there a simple way for the user to correct it?**
>
> Thanks for raising this valuable question. We performed an extra experiment to quantitatively assess their impact, by comparing the performance of DragFlow under **three** scenarios:
> - **(1) Null Prompt**;
> - **(2) Incorrect Prompt** (intentionally misinterpreted prompts);
> - **(3) Matched Prompt** (generated by MLLMs w.r.t. image and operation inputs).
>
> | **DragFlow w/**               | **$\text{IF}_\text{bg}$ ↑** | **$\text{IF}_\text{s2t}$ ↑** | **$\text{IF}_\text{s2s}$ ↓** | **$\text{MD}_\text{1}$ ↓** | **$\text{MD}_\text{2}$ ↓** |
> |-------------------------------|---------|---------|---------|---------|--------|
> | Null Prompt                     | 0.966   | 0.944   | 0.948   | 33.81   | 6.81   |
> | Incorrect Prompt              | 0.955   | 0.936   | 0.955   | 36.87   | 7.73   |
> | Matched Prompt (by QWen-VL)   | 0.968   | 0.947   | 0.945   | 32.42   | 6.25   |
> | **Matched Prompt (by GPT-5)** | **0.969**   | **0.948**   | **0.941**   | **31.59**   | **5.93**   |
>
> The averaged results on *DragBench-DR* above confirm our findings that operating with **"Null Prompt" leads to only a slight decrease** in performance metrics. This indicates that our framework is sufficiently robust on its own to handle the primary editing task. In contrast, using the **"Incorrect Prompt" shows a more negative impact**, as it may cause conflicting guidance.
>
> Furthermore, **"Matched prompt" yields better effectiveness than "Null Prompts"**. To explore the robustness, we examine matched prompts from two distinct well-known MLLMs. The outcomes indicate they are comparable, where **the impact of MLLM choice was not significant**.
>
> Regarding potential misinterpretations, users can **choose the most suitable option from several generated candidates before the editing begins**, rather than relying on post-hoc adjustments. This pre-execution “double confirmation” step ensures that the subsequent editing is guided strictly by the user’s precise intent, thereby minimizing ambiguity to the greatest extent.
>
> We have incorporated these discussions into ***Appendix D.3*** and ***D.4***. Please let us know if you have any further questions. Your feedback is truly appreciated and has been very helpful in improving our work.
>
> ---
>
> **[Q2] The framework currently handles relocation, deformation, and rotation via affine transforms. How would it generalize to more complex, non-affine warps like twisting or bending?**
>
> Thank you for this thoughtful question. In our framework, the currently implemented operations are intended to serve as **flexible building blocks**. They can be **composed to approximate a wide range of non-rigid, non-affine effects**. For instance, a "bending" transformation can often be represented as a sequence of localized rotations. We provide illustrative examples of these composite effects in ***Figure 14***.
>
> For more complex transformations such as "twisting", we acknowledge that this remains a challenging task for drag-based editors. It is inherently difficult for users to **precisely specify a 3D twisting operation using only 2D spatial controls**. In such cases, **text-driven editing** may offer a more natural and effective means of control.
>
> We sincerely appreciate your feedback, and we would be happy to provide further clarification.

---

### Official Review · Reviewer_EW7a · 2025-10-31

**Soundness:** 3
**Presentation:** 3
**Contribution:** 3
**Rating:** 6
**Confidence:** 4

**Summary:**

This paper introduces DragFlow, a novel framework for region-based drag editing tailored to Diffusion Transformers, addressing the limitations of point-based methods when applied to models like FLUX. The work first provides a detailed analysis of why point-based drag fails on DiTs and motivates the first region-based approach. DragFlow incorporates region-level affine supervision, gradient mask-based background preservation, and adapter-enhanced inversion to improve subject consistency. It also leverages MLLMs for intent inference and introduces the ReD Bench benchmark. Experiments demonstrate outstanding performance.

**Strengths:**

1. The paper thoroughly investigates the failure of point-based drag editing on DiTs, contrasting feature granularity between UNets and DiTs. This analysis provides valuable insights into model architectures and justifies the shift to region-based supervision. The motivation is strong and clear, as it addresses a critical gap in adapting drag editing to modern DiT-based models like FLUX.
2. DragFlow introduces a well-designed pipeline that replaces point-based supervision with region-level affine transformations, avoiding the need for explicit tracking and enhancing feature guidance. Together with gradient masks and pre-trained adapters to enhance background preservation and subject consistency.
3. Valuable ReD Bench benchmark with detailed annotations and demonstrates SOTA performance on both ReD Bench and DragBench-DR across multiple  metrics.
4. Effectively uses multimodal LLMs to interpret user intents and generate editing prompts, significantly reducing user interaction burden.

**Weaknesses:**

1. The paper lacks detailed statistics on time and memory consumption, which is critical given the optimization-based nature of DragFlow. With multiple iterations, inversion steps, and adapter integrations, the method likely incurs higher computational costs. A comparison of inference time and GPU memory usage would provide practical insights for real-world applications.
2. While DragFlow is validated on FLUX, its adaptability to other DiT-based models (e.g., SD3) is not explored. The paper does not address potential architectural or training differences that might affect performance, leaving questions about generalization.

**Questions:**

1. How essential are the MLLM-generated editing prompts? Would the method achieve comparable results without editing prompts?

---

> ### Author Response · Authors · 2025-11-24
> **Response to Reviewer EW7a (1/2)**
>
> Dear Reviewer EW7a,
>
> We sincerely appreciate your constructive review. You took the time to offer many valuable comments that contribute significantly to improving our work.
>
> We are particularly encouraged that you recognized our "thorough investigation" of DiT feature granularity as a "strong and clear" motivation. We also deeply appreciate your validation of our "well-designed pipeline", the contribution of our "valuable ReD Bench", and the practical effectiveness of our MLLM integration.
>
> We believe there may be a few misunderstandings, and we are glad to address your concerns point by point.
>
> ---
>
> **[W1] The paper lacks detailed statistics on time and memory consumption, which is critical given the optimization-based nature of DragFlow. With multiple iterations, inversion steps, and adapter integrations, the method likely incurs higher computational costs. A comparison of inference time and GPU memory usage would provide practical insights for real-world applications.**
>
> We sincerely thank you for reminding this. The comparison of wall-clock runtime and memory is summarized in the table below. The results are averaged on *DragBench-DR* using NVIDIA H100 GPU. We have included this information in ***Appendix I (Skyblue)***.
>
> |Methods|Prep + Edit Time (s)|Peak Memory (GB)|
> |-|-|-|
> |RegionDrag|9.2|4.3|
> |FastDrag|5.4|5.2|
> |InstantDrag|1.2|4.1|
> |DragLoRA|58.6|14.6|
> |FreeDrag|113.2|13.2|
> |DragNoise|103.4|12.2|
> |GoodDrag|104.6|12.8|
> |CLIPDrag|101.0|19.4|
> |DragDiffusion|82.1|12.1|
> |DragFlow|158.7|23.5|
>
> As the data present, DragFlow is **slower and more memory-consuming than methods based on much smaller SD models**. This is an expected result, as DragFlow is **the first one designed to leverage the significantly larger and more powerful DiT base models**.
>
> In our implementation, we use **qint8 quantization** to accelerate inference and reduce memory usage, and pair it with **CPU offloading** to further decrease GPU memory requirements. Beyond the current level, we believe there is still plenty of room to further reduce memory costs and increase speed through future improvements in both the algorithm and the code.
>
> ---
>
> **[W2] While DragFlow is validated on FLUX, its adaptability to other DiT-based models (e.g., SD3) is not explored. The paper does not address potential architectural or training differences that might affect performance, leaving questions about generalization.**
>
> We appreciate this suggestion regarding our framework generalization. To address this, we have conducted an additional experiment applying DragFlow to SD3.5. The details can be found in ***Appendix J.1 (Cyan)***.
>
> The averaged results on *DragBench-DR* are presented below. Notably, since the [*InstantCharacter-IP-Adapter*](https://huggingface.co/spaces/InstantX/InstantCharacter) is designed specifically for FLUX, we employed [*SD3.5-Large-IP-Adapter*](https://huggingface.co/InstantX/SD3.5-Large-IP-Adapter) as a comparable module for the SD3.5 edition.
>
> | **Methods** | **$\text{IF}_\text{bg}$ ↑** | **$\text{IF}_\text{s2t}$ ↑** | **$\text{IF}_\text{s2s}$ ↓** | **$\text{MD}_\text{1}$ ↓** | **$\text{MD}_\text{2}$ ↓**  |
> |------------------------------------------|-------|-------|-------|-------|------|
> | DragFlow (based on SD3.5) | 0.962 | 0.945 | 0.952 | 35.21 | 6.58 |
> | **DragFlow (based on FLUX.1-dev)**       | 0.969 | 0.948 | 0.941 | 31.59 | 5.93 |
>
> Our framework **retains good performance when transferred to SD3.5**. The FLUX edition shows a slight advantage, due to its richer generative prior, yet the overall performance on SD3.5 remains highly competitive. These results confirm that our framework is generalizable.

---

> ### Author Response · Authors · 2025-11-24
> **Response to Reviewer EW7a (2/2)**
>
> **[Q1] How essential are the MLLM-generated editing prompts? Would the method achieve comparable results without editing prompts?**
>
> Thanks for raising this valuable question. We performed an extra experiment to quantitatively assess their impact, by comparing the performance of DragFlow under **three** scenarios:
> - **(1) Null Prompt**;
> - **(2) Incorrect Prompt** (intentionally misinterpreted prompts);
> - **(3) Matched Prompt** (generated by MLLMs w.r.t. image and operation inputs).
>
> | **DragFlow w/**               | **$\text{IF}_\text{bg}$ ↑** | **$\text{IF}_\text{s2t}$ ↑** | **$\text{IF}_\text{s2s}$ ↓** | **$\text{MD}_\text{1}$ ↓** | **$\text{MD}_\text{2}$ ↓**  |
> |-------------------------------|---------|---------|---------|---------|--------|
> | Null Prompt                     | 0.966   | 0.944   | 0.948   | 33.81   | 6.81   |
> | Incorrect Prompt              | 0.955   | 0.936   | 0.955   | 36.87   | 7.73   |
> | Matched Prompt (by QWen-VL)   | 0.968   | 0.947   | 0.945   | 32.42   | 6.25   |
> | **Matched Prompt (by GPT-5)** | 0.969   | 0.948   | 0.941   | 31.59   | 5.93   |
>
> The averaged results on *DragBench-DR* above confirm our findings that operating with **"Null Prompt" leads to only a slight decrease** in performance metrics. This indicates that our framework is sufficiently robust on its own to handle the primary editing task. In contrast, using the **"Incorrect Prompt" shows a negative impact**, as it introduces conflicting guidance.
>
> Furthermore, **"Matched prompt" yields better effectiveness than "Null Prompts"**. To explore the robustness, we examine matched prompts from two distinct well-known MLLMs. The outcomes indicate they are comparable, where **the impact of MLLM choice was not significant**.
>
> We have incorporated these discussions into ***Appendix D.4 (Skyblue)***. Please let us know if you have any additional questions. Your feedback is genuinely helpful in strengthening our work.

---

### Official Review · Reviewer_mi22 · 2025-10-31

**Soundness:** 3
**Presentation:** 2
**Contribution:** 3
**Rating:** 6
**Confidence:** 4

**Summary:**

This paper addresses the challenge of applying drag-based image editing, a paradigm popularized by models like DragGAN, to modern generative models based on Diffusion Transformers (DiTs), such as FLUX.

**Strengths:**

1. The experiments are very sufficient, compared with almost all mainstream baselines, and the results are very convincing.
2. The proposed method sounds novel.  Similar to RegionDrag, this method doesn't need Point Tracking, which is very promising compared with previous drag-edit methods.

**Weaknesses:**

1. Motivation for some proposed components is not clearly explained. For details, see the questions part.

2.  Lack an introduction to the "drag edit" method, like the meaning of motion supervision, tracking. It is unrealistic to assume that all readers are familiar with drag edit.

3. The discussion about ReD Bench is insufficient. Since the authors claim ReD Bench as one of the contributions, I think more details should be provided (such as how the editing instructions and corresponding ground truth are generated, the dataset size, or even some examples).

**Questions:**

1.  Since the input for point-based and region-based editing is different, how can you guarantee the fairness of the comparison?

2.  I am confused about the statement in Lines 79-80. What is inversion drift? And why will CFG-distilled exacerbate it? Then why does inversion drift make KV injection insufficient?

3.  In this paper, only three kinds of operation are defined (relocation, deformation, and rotation).  Can they represent all the drag instructions? What if there is an instruction consists of a rotation and relocation?

4.  Lines 307- Line 309. About the experimental setting, it is quite rare in deep learning with a learning rate of 1000 or 1200. What's the reason?

---

> ### Author Response · Authors · 2025-11-24
> **Response to Reviewer mi22 (1/3)**
>
> Dear Reviewer mi22,
>
> Thank you very much for taking the time to review our manuscript and for offering those thoughtful comments that helped us improve the work.
>
> We greatly appreciate your recognition that our experiments are "very sufficient" and the results "very convincing." We also sincerely value your assessment of our method as novel and "very promising," particularly highlighting its advantage of eliminating the need for Point Tracking.
>
> We will address your concerns point by point:
>
> ---
>
> **[W2] Lack an introduction to the "drag edit" method, like the meaning of motion supervision, tracking. It is unrealistic to assume that all readers are familiar with drag edit.**
>
> Thank you for this valuable advice. The preliminary information and background for the "drag edit" task is included in ***Appendix B.2 (Darkred)*** for readers seeking a deeper review of the technical development track.
>
>
> ---
> **[W3] The discussion about ReD Bench is insufficient. Since the authors claim ReD Bench as one of the contributions, I think more details should be provided (such as how the editing instructions and corresponding ground truth are generated, the dataset size, or even some examples).**
>
> We agree with your comment. To address this, we have included a detailed description of the *ReD Bench* in ***Appendix G.1 (Darkred)***. This section now provides details on its construction, including the set size and instruction formation. As a preview, **two samples** are demonstrated in ***Appendix G.3 (Darkred)***.
>
> Following the design of previous drag-editing datasets [1,2], our proposed data bench **does not contain "ground truth" results**, which only provides multi-format instructions.
>
> Thank you again for your helpful feedback, which has allowed us to further improve the completeness and clarity of our work.
>
> [1] Shi, Y., Xue, C., Liew, J. H., Pan, J., Yan, H., Zhang, W., ... & Bai, S. (2024). Dragdiffusion: Harnessing diffusion models for interactive point-based image editing. In Proceedings of the IEEE/CVF Conference on Computer Vision and Pattern Recognition (pp. 8839-8849).
>
> [2] Lu, J., Li, X., & Han, K. (2024, September). Regiondrag: Fast region-based image editing with diffusion models. In European Conference on Computer Vision (pp. 231-246). Cham: Springer Nature Switzerland.
>
> ---
>
> **[Q1] Since the input for point-based and region-based editing is different, how can you guarantee the fairness of the comparison?**
>
> Thank you very much for this thoughtful question.
>
> 1. We agree that achieving a universally accepted notion of fairness between point-based and region-based editing is inherently challenging. Region-based inputs generally contain **less ambiguity** than point-based inputs, which is precisely part of the motivation behind the development of region-based methods.
>
> 2. Since region-based approaches emerged more recently and have fewer established baselines, we also include comparisons with point-based methods to provide readers with a **more comprehensive evaluation**. However, it is worth noting that our method not only outperforms point-based approaches, but also achieves superior results compared with region-based ones (see ***Table 2***).
>
> 3. On the other hand, regarding the experimental settings, we carefully ensure that all methods follow **the same editing intent, regardless of input format**. For both benchmarks, different input types are **aligned under rigorous human supervision** to maintain consistency in task objectives.
>
> We hope this addresses your concerns, and we sincerely appreciate your feedback. If you have any additional suggestions, we would be very happy to continue the discussion.

---

> ### Author Response · Authors · 2025-11-24
> **Response to Reviewer mi22 (2/3)**
>
> **[Q2] I am confused about the statement in Lines 79-80. What is inversion drift? And why will CFG-distilled exacerbate it? Then why does inversion drift make KV injection insufficient?**
>
> Thank you very much for the thoughtful question.
>
> 1. **What is inversion drift?**
>
>    “Inversion drift” refers to the **reconstruction error** that occurs when the latent obtained through inversion cannot be perfectly denoised back into the original image. In other words, the forward (inversion) and backward (reconstruction) processes do not align exactly.
>
> 2. **Why does CFG distillation exacerbate inversion drift?**
>
>    In diffusion models, accurate inversion and reconstruction rely on the quality of the **ODE approximation**. A precise vector field is crucial for ensuring that the ODE trajectory is reversible.
>
>    In the *non-distilled* setting, the unconditional output typically allows relatively accurate inversion [1]. However, introducing **classifier-free guidance (CFG)** perturbs the vector field used during inversion, reducing its accuracy [2]. Many prior methods therefore adjust the unconditional or conditional predictions to improve the inversion accuracy of the resulting CFG prediction [1-5], but these adjustments generally incur additional computational costs to maintain ODE reversibility.
>
>    In contrast, in a **CFG-distilled model**, the model no longer explicitly separates unconditional and conditional predictions. Instead, CFG is merged into a single numerical input. Without any intervention, this leads to less accurate model predictions and consequently poorer inversion and reconstruction quality (as shown in ***Table 1***).
>
>    For **step-distilled models**, the ODE trajectory becomes even more distorted, to the point where reversibility is almost entirely lost.
>
> 3. **Why does inversion drift make KV injection insufficient?**
>
>    KV injection aims to improve prediction accuracy by reusing cached features. However, when the underlying trajectory deviates significantly due to inversion drift, relying on features extracted from a flawed inversion becomes **insufficient**.
>
>    We find that using stronger representations of the subject (e.g., features extracted via an IP-Adapter) provides the model with a more reliable signal. This could lead to more accurate predictions and significantly improves inversion performance under CFG distillation.
>
> We hope this explanation is helpful, and we sincerely appreciate your constructive feedback. Please feel free to let us know if you have further questions. We would be more than happy to continue the discussion.
>
> [1] Lu, S., Liu, Y., & Kong, A. W. K. (2023). Tf-icon: Diffusion-based training-free cross-domain image composition. In Proceedings of the IEEE/CVF International Conference on Computer Vision (pp. 2294-2305).
>
> [2] Wallace, B., Gokul, A., & Naik, N. (2023). Edict: Exact diffusion inversion via coupled transformations. In Proceedings of the IEEE/CVF Conference on Computer Vision and Pattern Recognition (pp. 22532-22541).
>
> [3] Mokady, R., Hertz, A., Aberman, K., Pritch, Y., & Cohen-Or, D. (2023). Null-text inversion for editing real images using guided diffusion models. In Proceedings of the IEEE/CVF conference on computer vision and pattern recognition (pp. 6038-6047).
>
> [4] Ju, X., Zeng, A., Bian, Y., Liu, S., & Xu, Q. PnP Inversion: Boosting Diffusion-based Editing with 3 Lines of Code. In The Twelfth International Conference on Learning Representations (ICLR).
>
> [5] Miyake, D., Iohara, A., Saito, Y., & Tanaka, T. (2025, February). Negative-prompt inversion: Fast image inversion for editing with text-guided diffusion models. In 2025 IEEE/CVF Winter Conference on Applications of Computer Vision (WACV) (pp. 2063-2072). IEEE.
>
>
> ---
> **[Q3] In this paper, only three kinds of operation are defined (relocation, deformation, and rotation). Can they represent all the drag instructions? What if there is an instruction consists of a rotation and relocation?**
>
> Thank you very much for this thoughtful question. In our framework, the three currently defined operations (relocation, deformation, and rotation) are designed as **flexible building blocks**. These primitives can be **composed to approximate a broad range of non-rigid and non-affine drag instructions**.
>
> As a result, more complex interactions, such as the “rotation plus relocation” example you mentioned, can be **naturally expressed by combining multiple basic operations**.
>
> To better illustrate this capability, we have added visual examples in the *last three* rows of ***Figure 14***, and we have expanded the discussion in ***Appendix H.2 (Violet)*** to help clarify this point for readers. We sincerely appreciate your constructive feedback.

---

> ### Author Response · Authors · 2025-11-24
> **Response to Reviewer mi22 (3/3)**
>
> **[Q4] Lines 307- Line 309. About the experimental setting, it is quite rare in deep learning with a learning rate of 1000 or 1200. What's the reason?**
>
> Thank you very much for your question. You are right that learning rates as large as 1000 or 1200 are uncommon in typical model training. However, in our case we are not training a network; instead, we are **optimizing a latent**.
>
> The specific value (e.g., 1000) was selected based on the **scale of the gradients** in our implementation. Because the gradients with respect to the latent codes were small (likely due to the masked loss used in our region-based supervision), a larger learning rate is required to achieve an effective update magnitude. This ensured that the latent code could **move sufficiently and converge** within a reasonable number of iterations.

---

### Author Response · Authors · 2025-12-02
**Outline of Our Rebuttal**

Dear Reviewer, ACs, SACs, and PCs,

Thank you for your time and dedication. We are fully aware of your significant workload this year and deeply appreciate your efforts to uphold the integrity and professionalism of the review process despite the challenges.

For your convenience, we provide a summary below to help streamline the discussion. We hope this assists you in **quickly navigating the key issues addressed** during the rebuttal.

**Strength Overview.** As the reviewers noted, our work provides a fundamental insight and strong motivation for DiT-based editing **(EW7a, yXpA, ZsfJ)** and introduces a novel and technically sound drag edit framework **(mi22, EW7a, yXpA)**. The approach achieves convincing state-of-the-art performance **(mi22, yXpA, ZsfJ)** and contributes a valuable new benchmark **(EW7a, yXpA)**.

**Rebuttal Outline.** We outline the key points from our rebuttal revisions (**related revisions are colored** in the paper) as follows:

---
### **1. For `Reviewer mi22`**
* **[W2 & W3]** We added extra background details and *ReD Bench* explanations with visual samples.
* **[W1 & Q1]** We justified the fairness of our comparison by ensuring consistent editing intent under rigorous human supervision.
* **[W1 & Q2]** We clarified the mechanism of "inversion drift" in CFG-distilled scerinaros and analyzed why it necessitates our ID-preservation design (KV Cache + IP-adapter).
* **[W1 & Q3]** We confirmed and demonstrated the composability of our operations for complex edits (e.g., composite edits).
* **[W1 & Q4]** We justified the optimization hyperparameters and explained the rationale behind them.
---
### **2. For `Reviewer EW7a`**
* **[W1]** We provided an efficiency evaluation and justified the potential for further improvement.
* **[W2]** We present the competitive performance when adapting DragFlow to other DiT backbones. We validated the generalizability of our framework.
* **[Q1]** We conducted experiments that demonstrate the effectiveness of MLLM-generated prompts, confirming that our translating design produces optimal results.
---
### **3. For `Reviewer yXpA`**
* **[W1]** We demonstrated that the dependency on external adapters is not strict, as replacing it with subject-specific LoRA also yields excellent results.
* **[W2]** Same point requested in **EW7a [W1]**.
* **[W3]** We discussed the feasibility of integrating dense motion fields.
* **[Q1]** Same point requested in **EW7a [Q1]**. We also explained how our anti-ambiguity workflow works.
* **[Q2]** We confirmed and demonstrated the composability of our operations for complex edits (e.g., bending).
---
### **4. For `Reviewer ZsfJ`**
* **[W1]** Same point requested in **EW7a [W1]**.
* **[W2]** We added failure cases regarding imprecise instructions.
* **[W3]** We clarified that our main contribution is the first use of personalization adapters to mitigate inversion drift.
* **[Q1]** We confirmed and demonstrated the composability of our operations for complex non-rigid operations (e.g., hair or cloth).
* **[Q2]** We conducted a layer-wise ablation study to validate our design choices.
* **[Minor Suggestions]** (1) We revised the abstract accordingly. (2) We provided quantitative evidence and confirmed. (3) We stated the dataset usage in ablation studies.
---
We have thoroughly checked and incorporated the concerns raised to enhance the clarity and completeness of our work. We hope this outline can facilitate your review process, and we remain deeply grateful for all your contributions to the ICLR community.

Sincerely,

The Authors

---

### Meta-Review · Area_Chair_77Lg · 2026-01-05

**Summary:**

Reviewers found the method well-motivated, technically sound, and empirically strong, with consistent state-of-the-art results across benchmarks. Earlier concerns about clarity, benchmark details, and runtime analysis were largely resolved in the rebuttal. Overall, the paper offers meaningful contributions and convincing validation.

**Reviewer Concerns:**

It appears that the reviewers’ concerns have been adequately addressed in the rebuttal.

**Reviewer Scores:**

Reviewer mi22: 6
Reviewer EW7a: 6
Reviewer yXpA: 8
Reviewer ZsfJ: 6

Overall, the reviewers’ ratings are quite positive, and the rebuttal has addressed most of their concerns. I expect the reviewers to maintain or slightly raise their original scores.

---

### Decision · Program_Chairs · 2026-01-26

Accept (Poster)